# Perceptual restoration fails to recover unconscious processing for smooth eye movements after occipital stroke

**Sunwoo Kwon[1,2,3], Berkeley K Fahrenthold[3,4], Matthew R Cavanaugh[3,4], Krystel R Huxlin[2,3,4†], Jude F Mitchell[2,3*†]**

[1]Herbert Wertheim School of Optometry and Vision Science, University of California, Berkeley, Berkeley, United States; [2]Department of Brain and Cognitive Sciences, University of Rochester, Rochester, United States; [3]Center for Visual Science, University of Rochester, Rochester, United States; [4]Flaum Eye Institute,University of Rochester, Rochester, United States

**Abstract** The visual pathways that guide actions do not necessarily mediate conscious perception. Patients with primary visual cortex (V1) damage lose conscious perception but often retain unconscious abilities (e.g. blindsight). Here, we asked if saccade accuracy and post-saccadic following responses (PFRs) that automatically track target motion upon saccade landing are retained when conscious perception is lost. We contrasted these behaviors in the blind and intact fields of 11 chronic V1-stroke patients, and in 8 visually intact controls. Saccade accuracy was relatively normal in all cases. Stroke patients also had normal PFR in their intact fields, but no PFR in their blind fields. Thus, V1 damage did not spare the unconscious visual processing necessary for automatic, post-saccadic smooth eye movements. Importantly, visual training that recovered motion perception in the blind field did not restore the PFR, suggesting a clear dissociation between pathways mediating perceptual restoration and automatic actions in the V1-damaged visual system.

**\*For correspondence:**
jmitch27@ur.rochester.edu

†These authors contributed equally to this work

## Editor's evaluation

In this unique and meticulous study, Kwon, Huxlin and Mitchell add yet another twist to the story of dissociation between perception and action in the human brain. They found a sort of reverse blindsight: occipital stroke patients who lost conscious vision in part of the visual field could regain the ability to discriminate stimulus motion but did not make a specific form of eye movement that typically follows the stimulus motion. This result has implications for theories of perception and motor control, and for neurological rehabilitation.

## Introduction

Human observers use eye movements to bring targets of interest to central vision for detailed inspection. For moving targets, they do so effortlessly, with a combination of rapid saccades and smooth eye movements. When observers acquire a moving target via a saccade, they can continue to track it with smooth eye movements that match the eye's velocity to the motion of the target (*Buizza and Schmid, 1986*; *Lisberger and Westbrook, 1985*; *Lisberger et al., 1987*; *Rashbass, 1961*; *Tychsen and Lisberger, 1986*). Those pursuit movements can show accurate velocity tracking, matched to the target from the moment of saccade landing, indicating motion integration and predictive planning for the target prior to the saccade (*Gardner and Lisberger, 2001*). Other smooth eye movements can

occur involuntarily, as in ocular following, where the eyes drift after a saccade in response to the onset of wide-field motion (*Gellman et al., 1990*; *Miles et al., 1986*).

Recently, we found that saccade planning to peripheral, static apertures containing motion involuntarily generates predictive, smooth eye movements at the saccade target, even when there are no task demands to follow the target's motion (*Kwon et al., 2019*). These involuntary smooth eye movements anticipate the post-saccadic motion in the target aperture, generating a low-gain, following response along the target's motion direction, which we named the 'post-saccadic following response' (PFR). The PFR appears to reflect automatic, unconscious visual processing that occurs during a saccade target's selection – that is, during pre-saccadic planning (*Kwon et al., 2019*). Previous studies of pre-saccadic attention have shown that perceptual enhancements for the saccade target are automatic and obligatory (*Deubel and Schneider, 1996*; *Kowler et al., 1995*; *Rolfs and Carrasco, 2012*; *Rolfs et al., 2011*), and involve selection of target features such as spatial frequency and orientation (*Li et al., 2016*), as well as motion features (*White et al., 2013*), which can drive smooth eye movements. Thus, the PFR may represent an automatic consequence of attentional selection for the motion target prior to the saccade; alternatively, it may play a role in priming the motor system for subsequent tracking of that target. Critically for the current study, the automatic selection of stimulus motion during pre-saccadic attention results in PFR, which we will use as a read-out of the motion processing for smooth eye movements at specific peripheral locations. Both voluntary and involuntary smooth eye movements are thought to rely on processing of stimulus motion mediated through neural pathways in the middle temporal (MT) area (*Bakst et al., 2017*; *Mustari et al., 2009*; *Newsome et al., 1985*; *Nuding et al., 2008*). Area MT receives strong cortical inputs, routed through primary visual cortex (area V1), but it also receives direct input from sub-cortical centers, which bypass V1 (*Glickstein et al., 1980*; *Maunsell and van Essen, 1983*; *Rodman et al., 1989*; *Sincich et al., 2004*; *Tamietto and Morrone, 2016*; *Ungerleider et al., 1984*; *Van Essen et al., 1981*). To what extent these different routes of information that are transferred to MT contribute to voluntary and involuntary, smooth eye movements, and to perception, remains to be fully elucidated. Notably, prior studies have suggested that motion pathways driving involuntary smooth eye movements differ from those mediating perception (*Glasser and Tadin, 2014*; *Price and Blum, 2014*; *Simoncini et al., 2012*; *Spering and Carrasco, 2012*; *Spering et al., 2011*). As such, the fact that MT receives inputs from sub-cortical centers and from other cortical areas (via V1) prompted the hypothesis that sub-cortical pathways to MT may support smooth eye movements while conscious visual motion perception relies predominantly on input routed via V1 (*Spering et al., 2011*). Damage to V1 as a result of unilateral occipital stroke offers a unique opportunity to test this hypothesis in humans. Indeed, unilateral V1-strokes cause a loss of conscious visual perception in the contralateral visual hemifield (*Smith, 1962*; *Teuber et al., 1960*), but sub-cortical projections to MT are generally spared. Importantly, these projections are thought to underlie the preservation of unconscious residual abilities such as blindsight (*Mazzi et al., 2019*; *Sanchez-Lopez et al., 2019*; *Tamietto and Morrone, 2016*; *Weiskrantz et al., 1974*). A key role of MT in blindsight has also been inferred from the particular stimulus properties needed to elicit blindsight: visual targets presented in the blind field have to be relatively large, coarse, moving, or flickering (*Weiskrantz et al., 1995*), containing high luminance contrasts, low spatial frequencies, and high temporal frequencies (*Sahraie et al., 2008*) – stimulus properties that elicit strong responses from MT neurons (*Born and Bradley, 2005*; *Movshon and Newsome, 1996*). However, the impact of V1 damage on unconscious, motion-dependent visual processes used to guide smooth eye movements, such as the PFR, has not been investigated.

Moreover, although visual training can restore conscious visual motion perception in parts of the blind field of V1-stroke patients (*Cavanaugh et al., 2019*; *Cavanaugh et al., 2015*; *Das et al., 2014*; *Elshout et al., 2016*; *Huxlin et al., 2009*; *Saionz et al., 2020*; *Vaina et al., 2014*), we do not know how effectively patients can use such restored percepts to guide actions. Here, we used a cued, saccade task to demonstrate remarkably preserved ability of trained, V1-stroke patients to correctly target motion-containing peripheral stimuli presented in both their intact and blind fields. However, by continuously tracking eye movements, we were also – for the first time – able to capture the impact of V1 damage on the PFR. Our unique data reveal a key role for V1 in this unconscious, automatic, oculomotor behavior. By the same token, they provide new insights into the likely neural pathways mediating restored, conscious, motion perception after V1 damage versus those involved in the predictive processing necessary for a normal PFR.

# Results

To investigate how V1 damage impacts unconscious motion processing for smooth eye movements, we contrasted the PFR for saccades made to motion stimuli placed in the intact and the blind fields of eleven cortically blind (CB) V1-stroke patients (*Figure 1*) after visual restoration training at some of these blind field locations.

We identified four optimal equi-eccentric testing locations in the intact and blind fields of each participant, assessed from Humphrey perimetry (*Figure 1*), and measured both the PFR and motion integration thresholds using random dot stimuli (*Figure 2A*) at these locations. Our main goal was to assess how PFR tracked with perception in blind and intact regions of the visual field, rather than specifically at trained, blind field locations. As such, because PFR testing locations had to be within a certain range of eccentricities, and mirror symmetric across the four visual field quadrants, they did not always fall directly on all trained, blind field location. The tight location and task specificity of training-induced global motion recovery in the blind field (*Huxlin et al., 2009*; *Das et al., 2014*; *Saionz et al., 2020*) allowed us to attain a large range of perceptual thresholds at blind field PFR test locations, against which to plot PFR gain.

Global motion perception at PFR tested locations was assessed by measuring normalized direction range (NDR) thresholds (*Table 1*; additional details in Materials and methods). In brief, at each location of interest, patients were asked to perform a two-alternative, direction discrimination (left versus right) task using random dot stimuli in which the range of dot directions was varied in a 3:1 staircase procedure, from 0° (coherent motion) to 360° (random motion). We computed direction range thresholds using all trials in a given session, by fitting a Weibull function, which defined the direction range level at which performance reached 75% correct. Direction range thresholds were then normalized to the maximum possible range of motion of 360° (see Materials and methods) such that they ranged from 0 (good integration performance) to 1 (poor integration performance, or unmeasurable threshold). All participants in our sample obtained NDR thresholds at or below 0.3 for their intact visual fields and the next increment in the staircase procedure fell at 0.4. Thus, we set the upper limit of the normal threshold range at 0.35 – half way between those values.

In the blind fields, depending on whether the tested locations overlapped with a trained location, performance was either at chance or had improved to measurable and sometimes near-normal NDR thresholds, assessed in each patient's own intact visual field (*Table 1*). The end result was a set of 14 blind field locations across 11 patients, where perceptual thresholds ranged from unmeasurable (reflecting inability to do the task) to near-normal.

Oculomotor behavior was measured using a cued-saccade task for motion stimuli at four peripheral locations (colored circles in *Figure 1*; *Figure 2B*). Of particular note, the motion direction inside the apertures was irrelevant to the task and subjects were not asked to track it or report it. Additionally, in half the trials, the motion stimulus disappeared during saccade flight to disambiguate the contributions of pre-saccadic motion processing to the PFR.

## Basic saccade behavior of V1-stroke participants – accuracy and latency

Stroke participants were generally able to use central spatial cues at fixation to plan saccades to peripheral aperture locations. We first quantified saccade accuracy in a binary fashion by whether saccade end-points fell within the diameter of the stimulus aperture. We then also assessed spatial accuracy by the mean absolute error in position of the saccade end-point from the aperture center. A saccade was labelled 'correct' when it fell less than 3.5° from the saccade target center within 90 ms of the eye leaving the fixation window. Stroke participants made slightly more correct saccades to targets in their intact field (mean 96.8% ± 3.7%) than to those in their blind field (mean 89.4% ± 15.5%), likely reflecting more reliable target identification in the intact field ($t_{42}$=2.5078, p=0.016, BF = 3.4414). However, when correctly selecting targets in their blind fields, there was high spatial accuracy of saccades to those targets, as measured by the location of the end-points relative to the aperture center. Specifically, stroke participants had a mean absolute landing error relative to the stimulus center of 1.59°±0.07° (SEM) in the intact field and 1.59°±0.08° (SEM) in the blind field, which was not significantly different ($t_{10}$=0.1946, p=0.8496, BF = 0.3024). The latency of saccades was also similar for blind and intact field targets (intact field latencies: 360±45 ms; blind field latencies: 382±57 ms; $t_{42}$=−1.3593, p=0.181, BF = 0.6472).

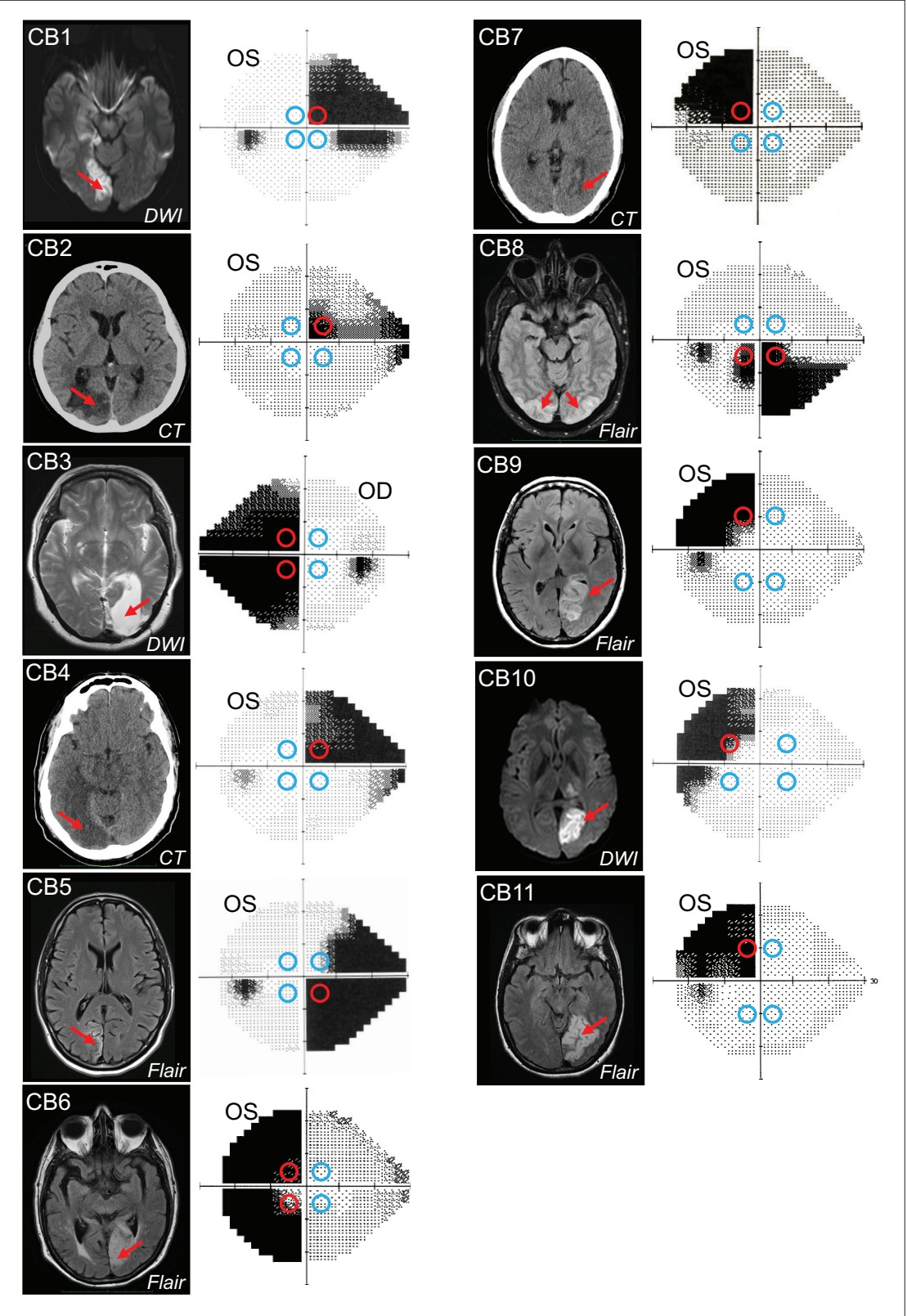

**Figure 1.** Occipital lesions, Humphrey visual field maps, and post-saccadic following response (PFR) testing locations. Single radiographic images through the brains of each V1-stroke participant (N=11), illustrating region(s) of occipital damage (red arrows), shown with right brain hemispheres on image right. The location, size, and shape of visual stimuli presented during the PFR testing protocol are indicated by colored circles superimposed on initial 24-2 Humphrey visual field maps acquired for the tested eye in each case. Red circles: blind field testing locations; blue circles: intact field

*Figure 1 continued on next page*

*Figure 1 continued*

locations; OS: left eye; OD: right eye; DWI: diffusion-weighted imaging; FLAIR: T2-weighted fluid-attenuated inversion recovery; CT: computed tomography.

Finally, we compared saccade accuracy and latency of stroke patients to those of visually intact controls from a previous study (*Kwon et al., 2019*). Saccade accuracies in the intact fields of stroke patients were slightly impaired compared to those of visually intact controls ($t_{36}$=1.5634, p=0.1267, BF = 0.8863) who exhibited saccade accuracy of 98.7% ± 0.8% as compared to 96.8% ± 3.7% for stroke patients. However, stroke patients had significantly longer ($t_{36}$=−5.4737, p<0.0001, BF > 1000) saccade latencies of 360±45 ms in their intact fields, compared to visually intact controls, whose latencies averaged 260±10 ms (*Kwon et al., 2019*). We consider several possible causes for these

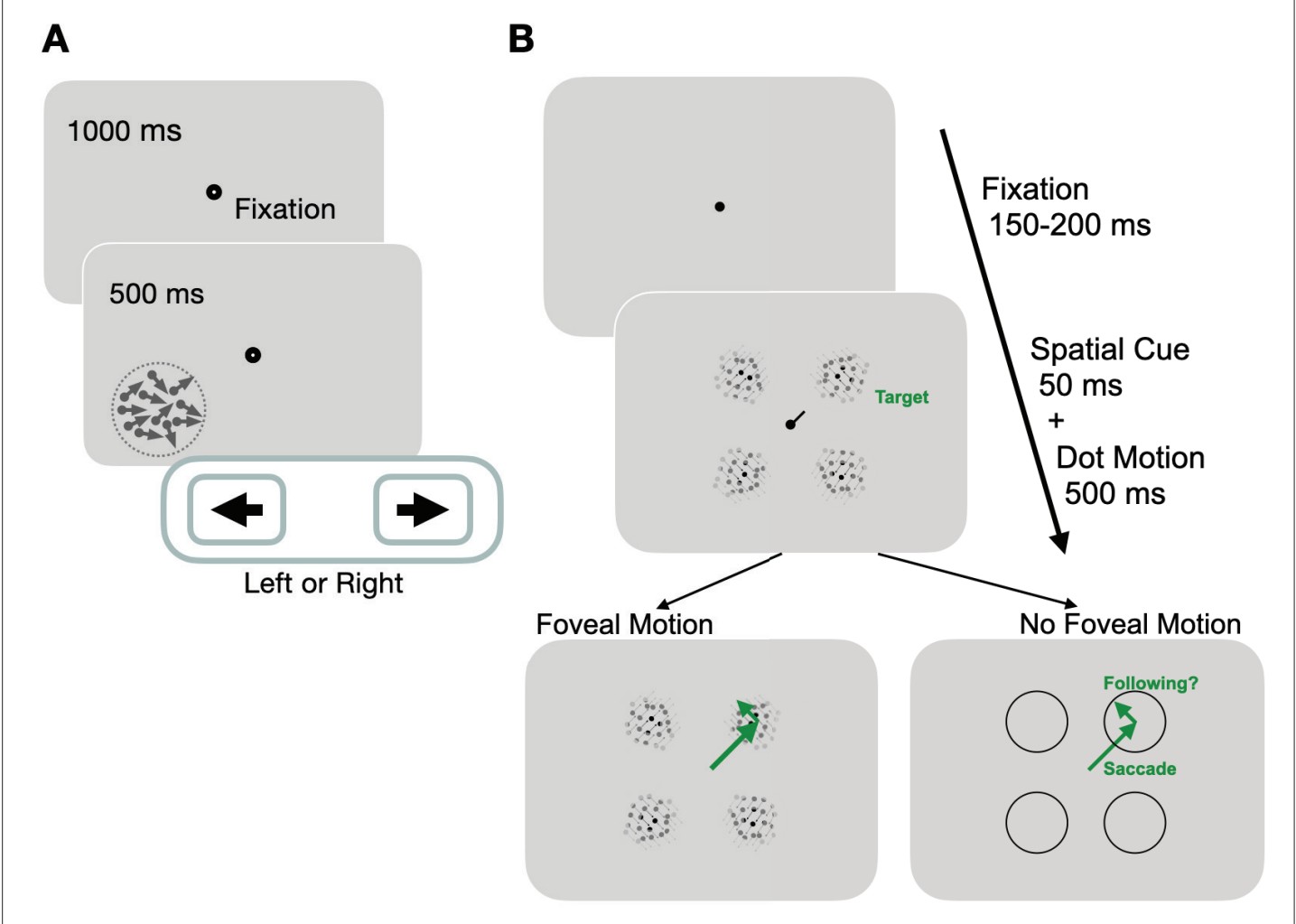

**Figure 2.** Experimental paradigms for measuring motion perception and oculomotor functions. (A) Trial sequence for assessing global motion perception: trials started with a fixation period of 1000 ms, followed by appearance of a random dot stimulus either in the blind or intact field for 500 ms. Dots moved globally to the right or left, with a range of directions defined an adaptive staircase. On each trial, subjects were asked to report the stimulus' global direction of motion by pressing the left or right arrow keys on a keyboard. They received auditory feedback on the correctness of each response. (B) Trial sequence for assessing oculomotor behavior: each trial started with a variable fixation period after which, participants were presented with four equi-eccentric motion apertures and a spatial cue at fixation. Dot motion apertures were Gaussian-enveloped and contained 100% coherent motion along a randomly assigned direction (clockwise or counter-clockwise) in each aperture, that was tangential to the center-out saccade to the aperture. The spatial cue (50 ms) indicated a peripheral target aperture toward which the participant was instructed to initiate a saccade as fast as possible. In half the trials, the dot motion stimuli persisted for 500 ms, and were thus present upon saccade offset. In remaining trials, stimuli disappeared during saccade flight, such that no stimulus motion was present at the fovea upon saccade landing.

**Table 1.** Demographics and global motion integration thresholds in retrained, V1-stroke participants.
M, male; F, female; NDR, normalized direction range (low value = good performance, 1 = bad performance/no measurable threshold). Each NDR threshold denotes performance measured at a single blind or intact field location in each patient (see *Figure 1* for positioning of these locations relative to the pre-training Humphrey visual field). Pre-training NDR thresholds were always 1 in blind fields.

| Subject | Sex | Age (years) | Time post-stroke (months) | Blind field NDR thresholds | Intact field NDR thresholds |
|---------|-----|-------------|---------------------------|----------------------------|-----------------------------|
| CB1 | F | 27 | 65.0 | 1 | 0.3 |
| CB2 | F | 68 | 24.8 | 0.2 | 0.2 |
| CB3 | F | 57 | 48.6 | 0.7, 0.8 | 0.3 |
| CB4 | M | 66 | 32.2 | 0.5 | 0.2, 0.2, 0.2 |
| CB5 | M | 54 | 52.2 | 1 | 0.3, 0.3 |
| CB6 | M | 79 | 22.7 | 1, 0.1 | 0.1, 0.2 |
| CB7 | M | 53 | 36.8 | 1 | 0.3 |
| CB8 | M | 52 | 65.5 | 1, 0.2 | 0.1, 0.2 |
| CB9 | F | 65 | 56.5 | 0.3 | 0.1, 0.1, 0.02 |
| CB10 | F | 31 | 66.8 | 0.3 | 0.2 |
| CB11 | F | 43 | 6.0 | 0.1 | 0.3, 0.2, 0.2 |

differences in the Discussion, including age, and challenges specific to saccade planning in the presence of a blind field.

## Predictive oculomotor behavior in the intact field of V1-stroke participants

In intact portions of their visual fields, stroke patients' post-saccadic smooth eye movements reflected the direction of target motion of the pre-saccadic stimulus immediately upon saccade offset. In a typical trial, a saccade made to a target aperture in the intact field exhibited a smooth drift in eye position from the saccade end-point along the direction of target motion (see example for a single stroke patient in *Figure 3A*). We quantified the time course of the drift in eye position by computing the eye velocity projected along the direction of target motion, where positive values reflect following of motion. We term this the PFR velocity. Across patients, we observed a net positive PFR velocity (*Figure 3C*). By including a stimulus manipulation in which the motion target disappeared during saccade flight, we were able to confirm that the PFR velocity was driven exclusively by pre-saccadic motion in the peripheral aperture. Within the first 100 ms after saccade offset (*the 'open-loop' period*) the PFR velocity did not differ depending on whether the stimulus remained present after the saccade (red trace in *Figure 3C*) or if it was removed during saccade flight (blue trace in *Figure 3C*). By including trials where we removed the stimulus in-flight, we could eliminate direct post-saccadic stimulation of motion at the fovea, and thus isolate the predictive, 'open-loop' component of the PFR velocity. After 100 ms from saccade offset, the presence or absence of foveal motion did influence post-saccadic eye movements. Specifically, the PFR velocity continued to increase along the target motion direction in the foveal-motion-present condition (red trace in *Figure 3C*), whereas it decreased when no stimulus was present upon saccade landing (blue trace in *Figure 3C*).

For subsequent analyses, we focused on the 'open-loop' period (i.e. within 100 ms after saccade offset), as the PFR velocity in this period depends on pre-saccadic motion information accumulated from the peripheral target aperture (*Kwon et al., 2019*). The average velocity of these open-loop following responses, both for stimulus-present and -absent trials, was small in magnitude relative to the stimulus speed (10 °/s), ranging from 5% to 15% relative velocity gain. These lower gain responses are consistent with an involuntary following response, such as ocular following (*Gellman et al., 1990*; *Miles et al., 1986*), rather than voluntary pursuit of a target. Overall, the pattern of oculomotor behavior in the intact field of stroke patients was highly similar in its time course and magnitude to that previously measured in visually intact controls (*Kwon et al., 2019*).

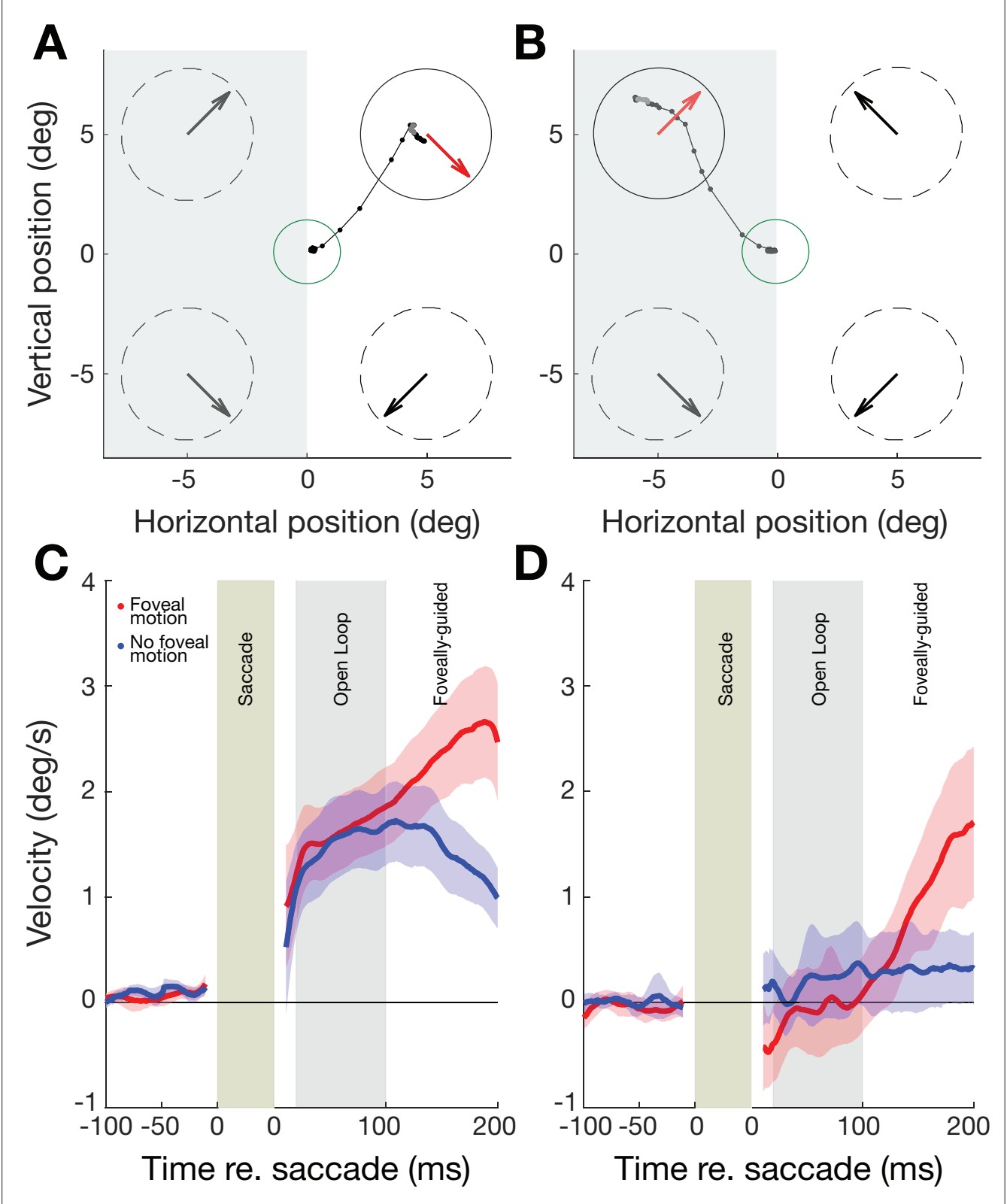

**Figure 3.** Oculomotor behavior in perceptually trained, 11 V1-stroke participants. (A) Eye movement traces to a cued target in the intact field (white background) of a single V1-stroke patient in a stimulus absent condition (i.e. with no foveal motion upon saccade landing at the cued target). Small, connected black dots denote the raw eye movement sampled from our eye tracker; the green circle represents the electronic window around the fixation spot; random dot stimuli were presented inside the four dashed circles and their global motion direction is indicated by large arrows inside

*Figure 3 continued on next page*

*Figure 3 continued*

each circle. Note the accurate saccade to the target center, and how the eye follows the pre-saccadic target motion direction (red arrow) upon saccade landing. (**B**) Raw eye movement traces to a cued target in the blind field (gray background) of a single V1-stroke patient in a stimulus absent condition. Note successful saccade landing onto the cued target but how the eye fails to follow the pre-saccadic target motion direction. Labelling conventions as in A. (**C**) Eye velocity traces for saccades to intact portions of the visual field, averaged across intact field locations per patient then averaged across all 11 stroke patients. In half the trials, stimuli were present upon saccade landing, resulting in foveal motion (red trace). In the remaining trials, stimuli were absent – that is, there was no foveal motion upon saccade landing on the target (blue trace). Error bars represent ± 2 SEM across subjects. Average eye velocities were projected along the target motion direction time-locked prior to the saccade onset (−100 to 0 ms) and offset (0 to 200 ms), such that positive values reflected motion consistent with the stimulus, while negative values reflected motion opposite. (**D**) Eye velocity traces for saccades to blind portions of the visual field, averaged across blind field locations per patient and then averaged across all 11 stroke patients (same conventions as in A). Error bars represent ± 2 SEM across subjects. Note the near-zero eye movement velocity during the 'open-loop' period, reflecting the lack of post-saccadic following response (PFR).

The online version of this article includes the following figure supplement(s) for figure 3:

**Figure supplement 1.** Post-saccadic following response (PFR) gain in the intact and blind field of V1-stroke patients during the foveally-guided period.

Visually intact controls in our prior study had a net positive PFR gain in the open-loop epoch that differed significantly from zero ($t_7$=4.31, p=0.004, BF = 14.2833 – dark gray bar in **Figure 4A**). PFR gain in the intact field of stroke patients (white bar in **Figure 4A**, 15.7% gain, $CI_{95}$ = [0.1564, 0.1576]) was not significantly different from PFR gain in our prior, visually intact controls ($t_{17}$=−0.577, p=0.5715, BF = 0.4578), and also showed a net positive effect that differed significantly from zero ($t_{10}$=5.4058, p<0.0001, BF > 100). Finally, we noted a significant correlation ($R^2$=0.351, $F_{18}$=9.73, p=0.0059, BF = 7.3164) between perceptual performance measured by NDR thresholds, and the magnitude of the PFR (white circles, **Figure 4B**) in the intact field of stroke participants.

## Predictive oculomotor behavior in the blind field of V1-stroke participants

Although saccades landed correctly on target stimuli in the blind field of our stroke participants ~89% of the time, post-saccadic eye movements differed dramatically from those in the same participants' intact field. Specifically, in the open-loop period, the eyes no longer moved along the direction of motion in the target (see example in **Figure 3B**) – in other words, there was no positive PFR velocity. This pattern was reflected in the average PFR velocity across all 11 stroke patients' blind fields (**Figure 3D**), irrespective of visual rehabilitation training. In contrast to oculomotor behavior when making a saccade to targets in their intact fields, stroke patients did not show any positive PFR velocity within the first 20–100 ms after saccade offset ('open-loop' period), whether the motion stimulus was present (red trace) or absent (blue trace) post-saccadically (**Figure 3D**).

Beyond 100 ms from the saccade offset, post-saccadic foveal motion – when present – was sufficient to drive an ocular following response (red trace, **Figure 3D**). The PFR during this latter foveally-guided period (100–200 ms) was driven by foveal motion for blind fields as much as the intact fields (**Figure 3—figure supplement 1**). PFR gain was positive and significant both for the intact fields (18.6% gain, $CI_{95}$ = [18.5%, 18.6%], $t_{10}$=4.2314, p=0.0017, BF = 25.4627) and the blind fields (15.0% gain, $CI_{95}$ = [14.9%, 15.0%], $t_{10}$=8.4130, p<0.0001, BF > 1000) with no significant difference between them ($t_{10}$=1.2041, p=0.2563, BF = 0.5355). This is an important observation, as it confirms that post-saccadic ocular following remained functional in these patients. Only the predictive component during the open-loop period was abnormal, reflected by the absence of the PFR velocity in stimulus-absent trials (blue trace, **Figure 3D**). Consistent with these observations, PFR gain in the blind field of our stroke patients (light gray bar in **Figure 4A**, 0.45% gain, $CI_{95}$ = [0.43%, 0.47%]) was significantly lower than PFR gain in their intact field ($t_{10}$=5.6451, p=0.0001, BF = 150), and was not significantly different from zero ($t_{10}$=0.3796, p=0.7122, BF = 0.3166).

Since portions of the blind fields of stroke patients underwent training that restored global motion perception, we next asked if such training, when it was coincident with locations tested on the PFR task, restored PFR gains. Perceptual recovery did not restore PFR: there was no significant correlation between NDR and PFR gains ($R^2$=0.0655, $F_{12}$=0.841, p=0.377, BF = 0.3020) in the blind field of stroke patients (gray circles, **Figure 4C**). At blind field locations where post-training stroke patients attained NDR thresholds < 1, the PFR gain was not significantly different from 0 (mean PFR gain = 0.0177 ± 0.07, $t_8$=0.7524, p=0.4734, BF = 0.4066). Even when we isolated blind field locations where training

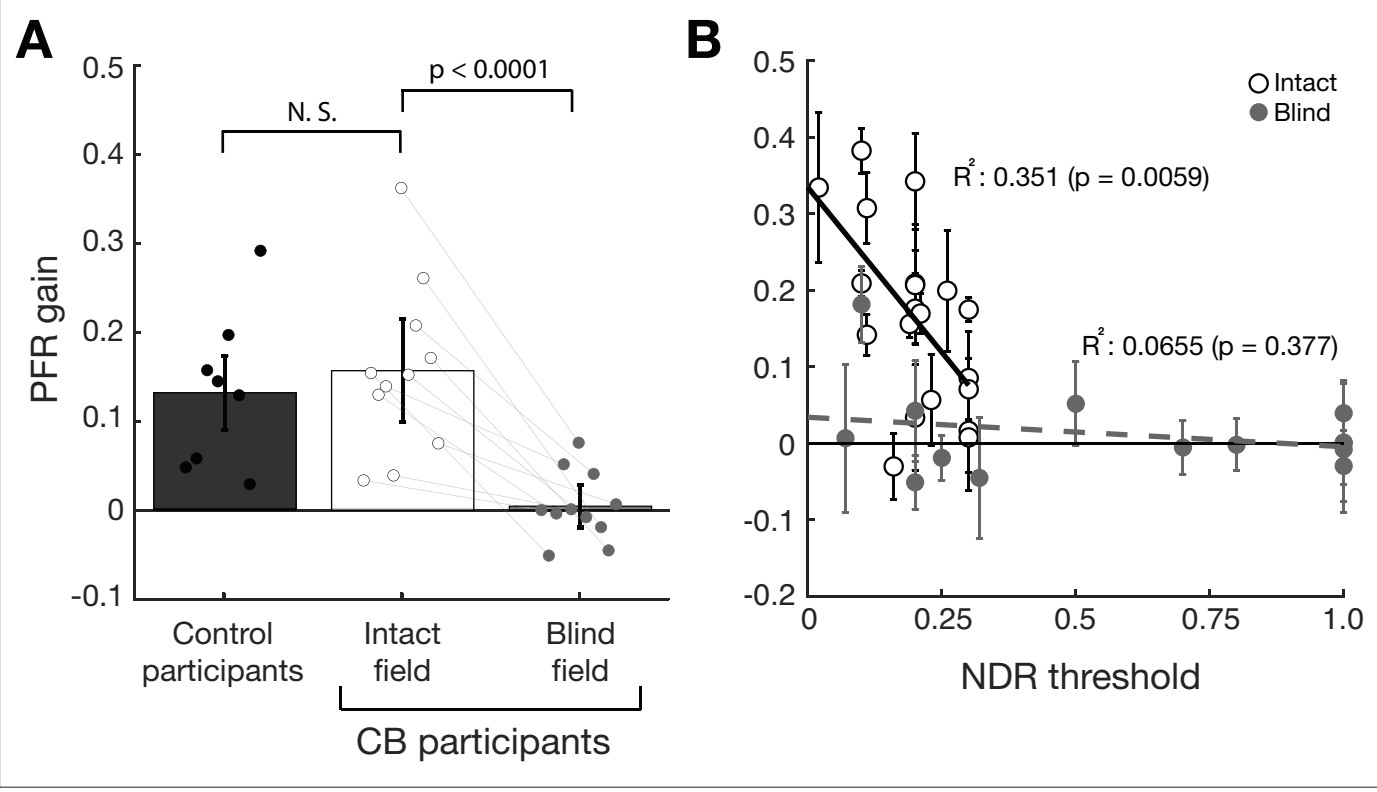

**Figure 4.** Post-saccadic following response (PFR) gain in the intact and blind field of V1-stroke participants. (A) Plot of mean PFR gain in eight visually intact controls (*Kwon et al., 2019*), and in 11 V1-stroke patients' intact and blind fields during the open-loop period. The PFR gain is represented as a proportion of pursuit gain along the target motion direction (1=perfect following, 0=no following, –1=following in the opposite direction). Individual dots represent the mean PFR for each participant. There was no significant difference in PFR gain between the intact fields of stroke patients and visually intact controls. PFR gain approximated 0 for saccades to targets in stroke patients' blind fields. Individual gray lines represent PFR gains in intact and blind fields belonging to the same patients. Each error bars represent ± 2 SEM across subjects. (B) PFR gain for individual intact field locations (open circles, N=20) and (C) blind field locations (gray circles, N=14) as a function of the normalized direction range (NDR) threshold measured at each location in stroke participants. The solid line and dashed line represent the line of best fit for intact and blind field PFR gain as a function of NDR threshold. Within intact regions of the visual field, there was a negative correlation between this global motion threshold and PFR gain: the lower the threshold (i.e. the better the motion perception), the higher the PFR gain. This relationship was lost in the blind field of stroke patients, with no significant correlation between NDR thresholds and PFR gain – the latter remaining abnormally low. Error bars represent 2 standard deviations per visual field locations.

The online version of this article includes the following figure supplement(s) for figure 4:

**Figure supplement 1.** Post-saccadic following response (PFR) gain as a function of normalized direction range (NDR) thresholds.

**Figure supplement 2.** Post-saccadic following response (PFR) gain as a function of normalized direction range (NDR) thresholds pooled per V1-stroke patients.

**Figure supplement 3.** Luminance contrast-dependence of the post-saccadic following response (PFR) gain in visually intact controls.

**Figure supplement 4.** Saccade angular deviations in the intact and blind field of cortically blind (CB) participants.

**Figure supplement 5.** Experimental paradigm for measuring oculomotor functions with a single saccade target.

**Figure supplement 6.** Post-saccadic following response (PFR) gain in the intact and blind field of V1-stroke patients driven by motion stimuli in the distractor apertures.

restored 'normal' NDR thresholds (<0.35), contrasting them against blind field locations where NDR thresholds were >0.35, PFR gain was not significantly different between those groups ($t_{12}$=−0.4140, p=0.6862, BF = 0.4749). Since many blind field locations failed to recover perception to the same range as the intact fields (NDR <0.35), we examined a subset of blind fields (N=6) where discrimination was matched to performance of intact fields. Still we found there was no significant PFR for those blind fields ($t_5$=0.5411, p=0.6117, BF = 0.4206) and that subset differed significantly from the intact fields ($t_{24}$=2.7424, p=0.0057, BF = 9.0755), and further the correlation remains non-significant for blind fields in that range ($R^2$=0.362, $F_4$=2.267, p=0.2066, BF = 0.6993). In an additional analysis, we

fit an exponential curve to describe how PFR varied with NDR for the intact fields, and asked if PFR at blind field location that recovered NDR (i.e. NDR <0.35) fell within its confidence intervals. Only one of six such blind field locations overlapped the intact field confidence intervals (*Figure 4—figure supplement 1*), providing strong evidence against those distributions being matched across NDR (p ≤ $1.79 \times 10^{-6}$, binomial test).

Finally, we should note that the above analyses treated individual visual field locations from the same participant as independent samples. Prior work in CB patients suggests that both baseline/pre-training performance and the ability of a location to improve are not uniform across blind fields, even at identical eccentricities; training-induced changes in discrimination performance are tightly restricted to the trained, blind field locations and do not spread by more than 1° outside the boundaries of each trained location (*Das et al., 2014*; *Huxlin et al., 2009*; *Sahraie et al., 2006*; *Saionz et al., 2020*). Nonetheless, we repeated our analyses by pooling the intact and blind fields within each participant and confirmed that our results remained consistent (*Figure 4—figure supplement 2*). In conclusion, over several tests, we found no significant recovery of PFR gain with recovery in perceptual motion performance.

## Discussion

For the first time, the present study measured the impact of V1 damage and subsequent, training-induced, visual restoration on both voluntary saccadic behavior and a class of unconscious, automatic, post-saccadic smooth eye movements: the PFR. Our findings revealed an unexpected, critical reliance of pre-saccadic visual motion processing on visual pathways that include V1. We first confirmed that V1-stroke patients exhibit normal saccade accuracy and normal PFR when saccading to motion targets in intact regions of their visual fields, where vision is mediated by intact V1. However, while the same patients exhibited normal saccade accuracies for targets presented in their blind fields, they had no measurable PFR, even after visual discrimination training recovered global motion perception at those blind field locations. Thus, restoration of motion perception did not automatically restore the unconscious visual motion processing necessary for the PFR. This is surprising because traditionally, patients with V1 damage are well known for having preserved, unconscious visual processing in their blind fields – under the umbrella of blindsight phenomena (reviewed in *Weiskrantz, 1996*; *Weiskrantz, 2009*). That visual discrimination training can recover the ability to perform the relatively complex computations needed to integrate motion direction into a global percept available to consciousness – even to the extent of attaining normal NDR thresholds – inside chronic blind fields is remarkable. That this could occur without automatically restoring the unconscious global motion processing necessary for predictive smooth eye movements was unexpected. Our findings suggest that primary visual cortex (V1) may be key for both conscious visual perception (*Tong, 2003*) and unconscious visual processes that influence smooth eye movements induced by peripherally presented motion targets. They also suggest that visual restoration, after V1 damage, recruits different neural circuits than are normally used for these processes in the intact visual system; finally, it suggests that these newly engaged circuits now dissociate conscious and unconscious visual motion processing.

The extrastriate visual area critical for many aspects of visual motion processing – area MT – receives strong inputs from V1 as well as from sub-cortical projections that bypass V1 (*Glickstein et al., 1980*; *Hagan et al., 2019*; *Maunsell and van Essen, 1983*; *Rodman et al., 1989*; *Sincich et al., 2004*; *Ungerleider et al., 1984*; *Van Essen et al., 1981*). This diversity of inputs to MT likely explains why, after V1 damage, residual visual motion processing persists inside the resulting blind fields (reviewed in *Das and Huxlin, 2010*; *Melnick et al., 2016*; *Tamietto and Morrone, 2016*). Key for rehabilitation efforts, this residual processing can be leveraged by intensive visual training to recover both simple and complex motion perception (*Cavanaugh et al., 2019*; *Das et al., 2014*; *Huxlin et al., 2009*; *Saionz et al., 2020*).

In non-human primates, activity in area MT has been causally linked to perceptual reports of motion in discrimination and detection tasks (*Britten et al., 1992*; *Newsome et al., 1985*; *Salzman et al., 1990*; *Siegel and Andersen, 1986*), and to accuracy in pursuit eye movements (*Huang and Lisberger, 2009*; *Newsome et al., 1985*; *Osborne et al., 2005*; *Salzman et al., 1990*; *Siegel and Andersen, 1986*). The perception of velocity is also well correlated with velocity gain in voluntary pursuit, supporting the notion that pursuit and perception share common motion processing at the neural level (*Gegenfurtner et al., 2003*; *Spering et al., 2005*; *Stone and Krauzlis, 2003*). Voluntary

pursuit and involuntary ocular following responses, such as the PFR, are also thought to rely on motion processing by the dLGN, V1, as well as area MT and the medial superior temporal (MST) area (*Bakst et al., 2017*; *Mustari et al., 2009*; *Nuding et al., 2008*; *Takemura et al., 2007*). Just as pursuit is modulated by stimulus contrast (*Spering and Gegenfurtner, 2007*; *Spering et al., 2005*), the PFR also exhibits a dependence on stimulus contrast; in visually intact humans, we saw a steep rise of the PFR contrast response function starting right below 10% luminance contrast, quickly reaching saturation at or above 15% contrast (*Figure 4—figure supplement 3*). This is comparable to the contrast response function of neurons in macaque area MT (*Heuer and Britten, 2002*; *Kohn and Movshon, 2003*; *Sclar et al., 1990*). However, pathways for ocular following are thought to be at least partly non-overlapping with those involved in motion perception (*Glasser and Tadin, 2014*; *Price and Blum, 2014*; *Simoncini et al., 2012*). Other studies in humans also support a distinction between neural circuits underlying smooth eye movements and conscious motion percepts (*Spering and Carrasco, 2012*; *Spering and Gegenfurtner, 2007*; *Spering et al., 2011*). Consistent with those studies, we now find that despite recovery of processing used for accurate, global motion perception in the blind field of V1-damaged humans, the PFR remains absent at trained, blind field locations.

One possible explanation for this outcome (disassociation between perception and PFR) is that training post-stroke lowered the thresholds used for reading out motion information in area MT for perception but not smooth eye movements. If motion perception in the blind field was consistent with weak residual sensory signals, and the PFR response has a higher contrast threshold than perception, then we could observe a lack of PFR while still retaining motion discrimination at blind field locations. Contrast thresholds for global motion stimuli in normal participants range between 5% and 10% contrast (*Fine et al., 2004*). We examined PFR as a function of contrast in normal participants to determine if it might have a higher threshold than that for perception, but found thresholds were in a comparable range (*Figure 4—figure supplement 3*). Thus, we consider it unlikely that the lack of PFR reflects a difference in threshold present in normal populations. However, there remains a possibility that perceptual training in patients may cause lower the threshold used for perception to accommodate weaker motion signals in area MT following V1 damage, but that a similar change in threshold did not transfer to the read-out used for oculomotor signals. This would reflect a disassociation between perception and eye movements at the level of read-out from area MT, which would still have important implications for visual recovery and understanding resulting differences in action and perception.

Another possible explanation for this disassociation is that while training post-stroke improved processing for perception, it did not correct problems with pre-saccadic attention and/or other aspects of saccade planning. It is well established that sensory processing among neurons in MT and MST can be strongly influenced by attention (*Treue and Maunsell, 1996*; *Treue and Trujillo, 1999*) and also by target selection immediately prior to saccades (*Ferrera and Lisberger, 1997*; *Recanzone and Wurtz, 2000*). Recent studies suggest that like target selection in voluntary pursuit, selective attention can modulate ocular following responses (*Souto and Kerzel, 2014*), and our findings here and previously (*Kwon et al., 2019*) support this notion. Pre-saccadic attention is thought to operate through feedback from oculomotor planning areas to visual cortex (*Moore and Armstrong, 2003*; *Moore and Fallah, 2004*), and while its impact has been studied mainly in visual area V4, it is also thought to occur in MT/MST. Indeed, electrical micro-stimulation in an oculomotor area, the frontal eye fields (FEF), influences selection of motion signals prior to saccades and can alter subsequent saccade trajectories to favor stimulus motion (*Schafer and Moore, 2007*). That the FEF and its projections to area MT are intact in V1-stroke patients suggests preservation of pre-saccadic planning and attention selection for the saccade target even when visual input is weak or abnormal in a blind field.

Although the effects of attention have not been studied extensively in V1-damaged patients, work to date suggests that some attentional mechanisms remain functional within cortical blind fields; as such, they could modulate motion signals at the level of MT/MST in the current behavioral paradigm. For instance, covert spatial attention was reported to improve stimulus detection in the blind field (*Poggel et al., 2006*) and in a separate study, it was shown to significantly decrease reaction times in V1-stroke patients performing an orientation discrimination task without any speed-accuracy trade-off (*Kentridge et al., 2004*). Feature-based attention was also able to improve fine direction discrimination training in cortically blinded fields (*Cavanaugh et al., 2019*). One piece of evidence suggesting that pre-saccadic attention remains functional in the current experiments is that other aspects of saccade pre-planning related to perceptual shifts in the position of motion targets remain

in the blind field. Previous studies reported a motion-induced perceptual shift for stimulus location along the direction of stimulus motion (*De Valois and De Valois, 1991*; *Nishida and Johnston, 1999*; *Ramachandran and Anstis, 1990*; *Whitney and Cavanagh, 2000*), which for saccades is reflected by a shift in their end-points along the direction of target motion (*Kosovicheva et al., 2014*; *Kwon et al., 2019*; *Schafer and Moore, 2007*). For stroke patients, we confirmed similar shifts in saccade end-points in their intact fields, as well as significant, albeit reduced shifts along the target motion in their blind fields (*Figure 4—figure supplement 4*). We also considered if the lack of PFR in the blind field may have resulted from inability to allocate attention for the saccade into the blind field, against the competing influence of distractor apertures within intact portions of the visual field. First, we ran a control experiment in two patients to compare PFR when competing distractors were absent (single aperture task) against the original design with four apertures (*Figure 4—figure supplement 5A*). A central line cue directed patients to the saccade target, and as in the main experiment, their saccades remained accurate for both intact (98.6% ± 1.2%) and blind field (98.7% ± 0.24%) locations. Importantly, we did not observe a measurable PFR in the blind fields when distractor motion was removed, and PFR remained robust in the intact fields comparable to the original design (*Figure 4—figure supplement 5B*). There also remained a correlation between PFR and NDR thresholds for intact fields, but not for the blind fields (*Figure 4—figure supplement 5C*). Thus, removing distractors had little impact on the PFR, suggesting the deficit was sensory in nature rather than involving a misallocation of attention. As a final control analysis, we also measured to what extent PFR was driven by the motion in distractor apertures. If the motion in the other three apertures produced any measurable influence on PFR, then one might expect that influence to be stronger when saccades were made into the blind field because attention could not be properly allocated to its location, resulting in higher attention to distractor locations. While we did find that motion in distractor apertures contributed to a weak PFR gain that was significant and measurable, contrary to that hypothesis, the gain was stronger for saccades made into intact visual fields as compared to blind fields (*Figure 4—figure supplement 6*). Thus, it seems unlikely that the lack of PFR reflects an impairment to engage pre-saccadic attention into the blind field. Instead, we posit that perceptual recovery through repetitive discrimination training did not entrain the specific motion processing pathways that support post-saccadic following.

An important consideration for the present experiments was whether failures to elicit PFRs in cortically blinded portions of the visual field might simply reflect a motor deficit for accurately targeting peripherally presented motion apertures. Because the PFR requires pre-saccadic attention to select the target motion, any loss in target localization accuracy could impair selection. Previous studies in monkeys with V1 lesions have found reduced spatial accuracy for saccades made into the blind field with larger end-point errors from 0.2° to 0.6° at matched eccentricities for intact and blind fields (estimated from data in Figure 2 of *Yoshida et al., 2008*). In the present study, we observed that stroke patients were less likely to correctly *select* a target aperture in their blind versus intact fields when given a central spatial cue (89% versus 97%). However, when the target was correctly selected, spatial accuracy of the saccade end-points was normal (1.31° versus 1.38° absolute error relative to the target's center). A key difference in the prior study is that *Yoshida et al., 2008*, used smaller stimuli, measuring only 0.45° in diameter, while we used large, dot motion fields (5.5° in diameter, Gaussian enveloped – see Materials and methods). Because spatial accuracy of saccade landings was similar for blind and intact fields of our stroke participants, we conclude that a motor deficit for targeting peripheral motion apertures in CB regions of the visual field was not a likely explanation for the absence of a PFR.

The slightly larger number of errors made by stroke patients when selecting cued targets in their blind fields could also be consistent with a reduction in their relative, perceived salience. Although all four motion apertures appeared simultaneously, iso-eccentrically and had equal (~100%) luminance contrast, it is possible that stroke participants required extra effort to ignore blind field-related perceptual inhomogeneities between the four apertures. Indeed, a prior study in chronic V1-stroke patients showed depressed luminance contrast sensitivity for motion and orientation discriminations at trained, blind field locations, in spite of normal NDR thresholds at these locations (*Das et al., 2014*). Thus, when patients are cued to saccade to a presumably less salient target in their blind field, this may require additional effort to suppress a reflexive saccade to the more salient targets. In an anti-saccade task where a salient target is ignored in order to plan a cued movement to the opposite (but empty) visual field, there is typically a reduction in saccadic reaction time (*Hallett, 1978*; *Munoz*

and Everling, 2004). If saccadic reaction times slow down by 140 ms or more due to task difficulty, this impacts saccades to both visible and anti-saccade locations due to the extra volitional demands (Hallett and Adams, 1980). In line with this observation, we found that saccade reaction times were slower for stroke patients by roughly 100–130 ms, for both the intact and blind fields, relative to visually intact controls. Controls did differ in age from stroke patients (~20 years versus ~57 years of age), and prior work showed a correlation between slower saccade latency and growing age; however, the typical reduction from 20 to 80 years of age was 40–45 ms (Abel et al., 1983; Pirozzolo and Hansch, 1981; Spooner et al., 1980). Therefore, it appears unlikely that age alone would account for the >100 ms reduction in saccadic reaction times in stroke patients. Rather, we posit that patients had to exert greater volitional control to select the cued target inside blind regions of their visual field.

In summary, V1 damage in humans, such as occurs from occipital stroke, causes a dramatic loss of conscious visual perception across large regions of the visual field, impairing most aspects of daily living. Paradoxically, this condition was famously known for its relative preservation of unconscious visual processes, such as those mediating blindsight. With the advent of visual restoration training for this patient population, an important question in the field has been to ascertain what aspects of visual processing can recover, which cannot, and why. Peripheral visual motion processing is key to many aspects of daily living. Not only is it critical for accurate perception and identification of targets, it is also essential for our motor actions and reactions to these targets. Here, we show that visual training that can restore perceptual discrimination of peripheral motion does not automatically recover the PFR (or normal saccade targeting to peripheral motion stimuli). Our findings support a dissociation between smooth eye movements, saccade targeting, and perception following V1 damage, and suggest that V1 is critical for driving smooth eye movements such as the PFR. A key realization emerging from these results is that alternative pathways, which convey motion information from sub-cortical centers directly to area MT, are insufficient to support predictive oculomotor behaviors when V1 is damaged, even if they are sufficient to mediate recovery of conscious motion perception.

A second insight attained presently is that repetitive motion discrimination training in CB fields might only influence circuits and processes that support conscious perception, without transfer to those driving motion-dependent, unconscious behaviors, such as the smooth eye movements involved in the PFR. Importantly, prior to global motion training, CB patients are typically unable to describe or even correctly guess global motion direction of random dot stimuli in their blind fields – as such, they have no demonstrable blindsight for the direction integration task used here (Das et al., 2014; Huxlin et al., 2009; Saionz et al., 2020), and for other global motion tasks (Azzopardi and Cowey, 2001; Vaina et al., 2014). By the end of training, CB patients in the present study could correctly describe random dot stimuli and attain relatively normal NDR threshold levels of performance – implying recovery of conscious perception and not simply more effective blindsight. A dissociation between processing for conscious vision and action in the context of restoring vision in CB naturally leads to parallels with the work of Goodale and Milner, 1992, which suggested that ventral stream projections from striate cortex to the inferotemporal cortex play a major role in perception, while dorsal stream projections from striate cortex to posterior parietal regions mediate the required sensorimotor transformations for visually guided actions. Our current results suggest that after V1 damage, psychophysical training enables residual motion processing, likely in area MT and related dorsal pathway areas, to be used to recover perception without recovering the necessary processing for smooth eye movements. This suggests a dissociation between perception and visually guided actions in this rehabilitation paradigm.

It remains to be determined if deliberate training on tasks that focus on saccade planning to motion targets might recover predictive motor behaviors. Rehabilitation of predictive ocular behaviors remains an uncharted area of research for V1-stroke patients, even though saccade training is one of the few forms of rehabilitation more readily available to these patients (Kerkhoff, 1999; Kerkhoff, 2000; Kerkhoff et al., 1992; Mannan et al., 2010; Nelles et al., 2001; Ong et al., 2015; Pambakian et al., 2004; Roth et al., 2009; Sahraie et al., 2016; Spitzyna et al., 2007; Trauzettel-Klosinski, 2010; Weinberg et al., 1977; Zihl, 1980). Of relevance to our observation of an apparent dissociation between perception and eye movements after V1 damage, training patients to saccade to targets in their blind field does not induce perceptual recovery (Campion et al., 1983; Pollock et al., 2019). Nonetheless, it is conceivable that an approach combining perceptual training with training of ocular

behaviors could improve the efficiency with which patients use information from their blind fields in everyday life.

## Materials and methods

### Participants

Eleven participants with long-standing cerebral blindness (CB) were recruited 2–5 years after a stroke that damaged their V1 unilaterally or in one case, bilaterally (see *Table 1* for details). The location and nature of V1 damage was verified from clinical brain imaging performed as part of each patient's standard of care. Homonymous visual field defects were confirmed using monocular, Humphrey automated perimetry performed at the Flaum Eye Institute of the University of Rochester (*Figure 1*). Participants suffering from neglect, cognitive impairments or ocular diseases were excluded from enrollment, as were those using psychoactive drugs. All participants had their visual acuity corrected to normal (with glasses or contact lenses) during testing.

Testing of V1-stroke participants occurred following completion of separate, visual training studies whereby they underwent visual training at one or more blind field locations. Some of these trained, blind field locations overlapped with PFR testing locations – some did not. The end result was a set of 14 blind field locations from 11 patients, where pre-training performance was initially at chance – that is, participants were unable to reliably discriminate left from right coherent, global motion. Post-training, however, depending on whether the tested locations overlapped with a trained location, performance either remained at chance or improved, generating measurable and occasionally near-normal direction integration (NDR) thresholds (*Table 1*). For comparison we also measured performance at a set of 13, iso-eccentric, intact field locations (*Table 1*; see below for details of global motion assessment methods).

PFR data from stroke patients were contrasted with a previously published data set obtained from eight visually intact controls (18–22 years of age; four females and four males) who had normal or corrected-to-normal vision (*Kwon et al., 2019*).

All procedures were approved by the Institutional Review Board of the University of Rochester and adhered to the tenets of the Declaration of Helsinki. Written, informed consent was obtained from each participant, and participation was at all times completely voluntary.

### Testing and visual training to recover perception

In each participant, visual field deficits were first estimated from Humphrey visual perimetry (*Cavanaugh and Huxlin, 2017*). This served as a starting point to map the position of the blind field border and select locations for PFR testing. It is important to note, however, that spared Humphrey visual field performance (i.e. luminance detection) was not necessarily representative of spared motion discrimination at a given blind field location – some patients were unable to attain normal global motion thresholds at locations partially spared on Humphrey perimetry (i.e. CB6 blind field location 2, *Figure 1*, *Table 1*, *Figure 4—figure supplement 1*). Blind field training was performed as previously described (*Cavanaugh et al., 2022*; *Cavanaugh et al., 2015*; *Das et al., 2014*; *Huxlin et al., 2009*; *Saionz et al., 2020*). In brief, participants were assigned to train daily at home using a custom MATLAB-based software (*Cavanaugh et al., 2021*) on their personal computers and displays. They were supplied with chin/forehead rests and instructed to position them such that the eyes were 42 cm away from their displays during training. They performed 300 trials per training location per day, at least 5 days per week, and emailed their auto-generated data log files back to the laboratory for analysis weekly. Session thresholds were calculated by fitting a Weibull function with a threshold criterion of 75% correct performance. Once performance reached levels comparable to equivalent intact field locations, training moved 1° deeper into the blind field along the x-axis (Cartesian coordinate space). When subjects returned to the lab, discrimination performance at all home-trained locations was verified with online fixation control using the Eyelink 1000 eye tracker. In addition, we measured NDR thresholds at blind and intact field locations chosen to undergo testing on the PFR task (see below). Intact field measurements were used to obtain a normative range of NDR thresholds that was specific to each participant (internal control for blind field performance). Ultimately, this enabled us to assess whether PFR was tied to perceptual motion integration performance in both intact and impaired regions of the visual field.

The locations used for PFR testing were mirror symmetric across the vertical and horizontal meridians, and ranged in eccentricity from 5.8° to 10.4° with a mean of 7.7°. This was done to maintain consistency with the original study (*Kwon et al., 2019*) that measured PFR in visually intact participants, whose stimuli uniformly sampled ordinal and cardinal directions at 7.1° eccentricity. Partly because of this experimental requirement, PFR testing locations did not always coincide with all trained locations in CB patients' blind fields. However, this was not a problem, as our main goal here was to ascertain if and how PFR gain tracked with NDR performance. Thus, we sampled regions of the blind field that underwent training and recovered normal NDR performance (CB2, CB6 location 2, CB8 location 2, CB9, CB10, CB11), as well as locations that were not directly trained or were trained on a non-global motion task, causing NDR thresholds to remain at 1 (CB1, CB5, CB6 location 1, CB7, CB8 location 1) or improve only partially (CB3 locations 1 and 2, CB4). In sum, the specificity of recovery to trained locations and tasks in CB fields (*Das et al., 2014*; *Huxlin et al., 2009*; *Sahraie et al., 2006*; *Saionz et al., 2020*) allowed us to attain a wide range of NDR threshold outcomes, against which to map PFR gain.

## Apparatus and eye tracking for assessing global motion perception

Participants were asked to perform 100 trials of a two-alternative, forced-choice, left-versus-right, global direction discrimination task at two to four, equi-eccentric, peripheral visual field locations chosen for testing of predictive oculomotor behavior (circles superimposed on Humphrey visual fields in *Figure 1*; red: blind field locations, blue: intact field locations). All blind field locations were tested in each patient (red circles in *Figure 1*). Time limitations restricted our ability to measure performance at every intact field location (blue circles in *Figure 1*), but at least one intact field location was assessed in each participant. Across intact field locations tested, we saw normal NDR thresholds that varied from 0.1 to 0.3 (*Table 1*). Percent correct and direction range thresholds were measured during in-lab testing, with central fixation enforced using an Eyelink 1000 eye tracker (SR Research, Mississagua, Ontario, Canada). Tracking was binocular for all participants except for CB3, who was tested monocularly because she exhibited convergence issues. As such, she had her dominant (right) eye tracked and the non-dominant eye patched both for motion perception and for PFR testing. Stimuli were presented in a gaze-contingent manner in either intact or blind regions of the visual field. Viewing distance to a luminance-calibrated CRT monitor (HP 7217A, 48.5 × 31.5 cm, 1024 × 640p, refresh rate 120 Hz) was 42 cm, enforced by a chin/forehead rest. Experiments were conducted using MATLAB (the MathWorks, Natick, MA) and the Psychophysics toolbox (*Brainard, 1997*; *Kleiner et al., 2007*; *Pelli, 1997*). At the start of each trial, subjects were asked to fixate a small target at the center of the CRT monitor. The Eyelink 1000 eye tracker was accurate to within 0.25°, with a sampling frequency of 1000 Hz (nominal, standard values as reported by SR Research). Subjects were allowed a fixation window of only ±1° around the fixation spot. If gaze moved outside this window during stimulus presentation, the trial was aborted, reshuffled and patients received a noxious auditory tone as feedback, reminding them to improve their fixation accuracy.

Following accurate fixation of the central spot for 1000 ms, a random dot stimulus appeared in a 5° diameter circular aperture, at one of the pre-determined locations in the peripheral visual field (see colored circles in *Figure 1*; NDR thresholds in *Table 1*). Black dots moved on a mid-gray background with a 250 ms lifetime, a speed of 10 °/s, and with a density of 3 dots/°². Stimuli were presented for 500 ms, accompanied by a tone to indicate stimulus onset. Dots moved globally with a variable range of directions, uniformly distributed around the left- or rightward vectors (*Das et al., 2014*; *Huxlin et al., 2009*; *Saionz et al., 2020*). On each trial, subjects were asked to report the stimulus' global direction of motion by pressing the left or right arrow keys on a keyboard (*Figure 2A*). Task difficulty was adjusted using an adaptive staircase (*Levitt, 1971*), which increased the range of dot directions from 0° to 360° in 40° steps after each set of three consecutive, correct responses; direction range was decreased by one 40° step for every incorrect response (*Das et al., 2014*; *Huxlin et al., 2009*; *Saionz et al., 2020*). Auditory feedback was provided on each trial, indicating the correctness of each response. Participants completed 100 trials at each of the four visual field locations. For each location tested, we fit a Weibull function to the data to extract the direction range threshold corresponding to 75% correct performance. This value was then normalized to the maximum possible range of dot directions (360°), generating an NDR threshold (*Das et al., 2014*; *Huxlin et al., 2009*), defined as:

$$\text{NDR threshold} = \frac{360^\circ - \text{Weibull fitted direction range threshold}}{360^\circ}$$

## Apparatus and eye tracking for PFR measurements

Stimuli were generated using the Psychophysics toolbox in MATLAB 2015b on a PC computer (Intel i7 CPU, Windows 7, 8 GB RAM, GeForce Ti graphics card). They were presented on a gamma corrected display (BenQ X2411z LED Monitor, resolution: 1920 × 1080p, refresh rate: 120 Hz, gamma correction: 2.2) which had a dynamic luminance range from 0.5 to 230 cd/m$^2$, at a distance of 95.25 cm in a dark room. Brightness on the display was set to 100% and contrast to 50%, and additional visual features of the monitor such as blur reduction and low blue light were turned off. Gamma corrections were verified with measurement by a photometer. Position of the left eye was recorded continuously in all participants except for CB3, who had her right eye tracked (see above). Eye position was recorded at 220 Hz using an infrared eye tracker (USB-220, Arrington Research, Scottsdale, AZ). The accuracy of the Arrington Eye Tracking system was 0.25°, with a precision of 0.15° (nominal, standard values as reported by Arrington Research). To minimize potential head movements, participants performed the task using a bite bar.

## PFR stimulus and task

CB patients performed a centrally cued saccade task toward peripheral motion apertures (**Figure 2B**) as previously described in visually intact controls (**Kwon et al., 2019**). In brief, and as schematically illustrated in **Figure 2B**, trials were initiated by fixation of a small, dark, fixation spot presented on a gray background. After a variable fixation period of 150–200 ms, a saccade cue appeared at fixation together with four dot motion apertures in a square configuration (colored circles, **Figure 1**). The cue (dark bar, 1° in length, extending from fixation) was used to indicate the target aperture to which the participant should saccade. The target aperture was 5.5° in diameter and was consistent for all the participants. For CB1, the target apertures were centered at (± 3°, ± 5°). For CB2-8, the target apertures were centered at (± 5°, ± 5°). For CB9, the target apertures were centered at (± 5°, ± 10°). For CB10, the target apertures were centered at (± 8°, ±5°). For CB11, the target apertures were centered at (± 3°, ± 10°). There were 180 dots total in each aperture, with dot luminance set to 0.5 cd/m$^2$ (100% contrast) and dot velocity fixed at 10 °/s. Following parameters from our previous study (**Kwon et al., 2019**), a Gaussian envelope was applied to each dot motion aperture to create a gradient in dot contrast from the center of the aperture (sigma = 1°).

To avoid stereotyped eye movements, we varied saccade directions across trials. Thus, the spatially cued motion aperture could appear in the intact or blind field of a given participant, on any given trial. Of particular note, *the motion itself or its direction were irrelevant to the task*. The motion within the aperture was 100% coherent and ran along a direction that was tangential to an imaginary line from the fixation point to the aperture. For each aperture, the motion was selected independent of the other apertures in one of the two tangential directions relative to the center-out saccade, either clockwise or counter-clockwise relative to the screen center.

We first compared eye movements in which the peripheral motion aperture was either present or absent upon saccade offset. Participants were instructed to make a saccade to the peripheral aperture as quickly as possible following the movement cue. A saccadic grace period (i.e. a maximum latency) was allowed for participants to initiate the saccade. In half the trials, selected at random, the stimulus motion remained present in all four apertures for 300 ms following detection of the eye landing within 3.5° from the center of an aperture. In the other half of trials, the stimulus was removed as soon as the eye had been detected leaving the fixation window, thus leaving a blank screen through the post-saccadic period. A saccade was labelled 'correct' when it fell at least 3.5° from the saccade target center within 90 ms of the eye leaving the fixation window. Participants completed as many trials as possible during a 1–1.5 hr session that sampled equally from the four locations and different stimulus conditions. During a typical session, stroke participants correctly completed on average 75.5 ± 32.0 (1 SD) trials in each blind field and 75.0 ± 31.8 (1 SD) trials in each intact field. For comparison against a set of eight normal controls, we analyzed eye movement data from a previous study (**Kwon et al., 2019**). The previous study employed a task with a subset of trials identical to the current study, but also in another half of trials tested an alternative stimulus configuration with the four apertures along the cardinal axes rather than the quadrants, thus a total of eight locations. During a typical session, normal controls completed on average 59.6 ± 13.8 (1 SD) trials in each of the eight fields.

## Eye movement recordings and PFR analysis

Eye position data were collected as participants performed saccades from fixation to the peripheral target. Eye tracking and saccade detection procedures were identical to those previously published (*Kwon et al., 2019*). We sub-sampled eye position using the ViewPoint Matlab toolbox (Arrington Research) at the display refresh rate (120 Hz) to initiate gaze-contingent task events. For offline detection of saccadic eye movements, we used the full eye position data recorded at 220 Hz and applied an automatic procedure that detected deviations in 2D horizontal and vertical eye velocity space (*Engbert and Mergenthaler, 2006*; *Kwon et al., 2019*). Only the trials where the saccade was labelled 'correct' were included in the PFR analysis. We then focused our analysis by time locking eye velocity traces on intervals 200 ms prior to saccade onset and 200 ms following saccade offset. Details for eye position filtering, smoothing, and saccade detection were as previously described (*Kwon et al., 2019*). In brief, the 2D eye velocity was computed from smoothed eye position traces and then projected onto the motion vector in the target aperture on each trial. These projected velocity traces were then aligned to saccade onset or offset, and averaged across trials for each participant. Notably, the stimulus motion in our task was selected such that it was tangent to the radial direction of the saccade from fixation out to the aperture (thus either clockwise or counter-clockwise directions of motion), as in the original study (*Kwon et al., 2019*). This minimizes the influence of velocity transients from the saccade itself in measuring the PFR. We have focused analyses on the eye velocity projected onto stimulus motion for this reason, as it provides a measure of pre-saccadic motion integration that is less contaminated by saccade-induced velocity. To quantify the net target-related eye velocity in each trial, we used a second measure of eye velocity that did not involve any filtering or smoothing of eye position. We computed a vector for the PFR in units of velocity (°/s) as the 2D vector difference in the raw (non-smoothed) eye position from 20 to 100 ms after saccade offset normalized by that time interval. Excluding the first 20 ms after saccade offset from this analysis interval reduced the influence of saccade-related effects to instead focus on post-saccadic smooth movements. Like velocity traces, we projected this 2D vector onto the vector of the target's motion to produce a single velocity value along the axis of stimulus motion, which we term the 'open-loop' PFR (*Kwon et al., 2019*). To assess the average PFR across trials, we computed each CB patients' eye movements relative to the target motion direction so that positive average eye velocities meant that the eye was moving along the target motion direction, and negative average eye velocities meant that the eye was moving opposite to the target motion direction.

Finally, we considered to what extent the PFR tracked target velocity by quantifying the PFR gain: the eye velocity computed from the open-loop PFR normalized to the target velocity, with +1 indicating a perfect match of eye velocity to the target motion, and negative values indicating eye velocity in the opposite direction.

## Statistics

To evaluate the significance of PFR gain we computed the one-sample t-test to verify it was greater than zero and we also computed the two-sample t-test to compare whether the PFR gains differed between conditions comparing either intact versus blind fields for CB participants, or intact fields for CB participants versus normal controls. We used the Pearson correlation to assess the relationship between the PFR gains and NDR thresholds within each stroke patient for intact and blind field visual locations. Confidence intervals of 95% were computed by bootstrapping the respective data at 10,000 times. Further statistical significance was computed by calculating the Bayes factor for the respective t-tests and ANOVA.

## Acknowledgements

The authors wish to thank Martin Rolfs, Duje Tadin, Ralf Haefner, and Gregory DeAngelis for enlightening discussions and comments on the manuscript. The authors also wish to thank Terrance Schaefer, who performed the Humphrey visual field tests presented here and Christine Callan for her excellent work as research coordinator. The present study was funded by NIH (EY027314 and EY021209 to KRH; EY030998 to JFM; P30 EY001319 to the Center for Visual Science), by start-up funds to JFM from the University of Rochester, and by an unrestricted grant from the Research to Prevent Blindness (RPB) Foundation to the Flaum Eye Institute.

# Additional information

### Competing interests
Krystel R Huxlin: co-inventor on US Patent No. 7,549,743. The other authors declare that no competing interests exist.

### Funding

| Funder | Grant reference number | Author |
|--------|------------------------|--------|
| National Eye Institute | EY027314 | Krystel R Huxlin |
| National Eye Institute | EY021209 | Krystel R Huxlin |
| National Eye Institute | EY030998 | Jude F Mitchell |
| Research to Prevent Blindness | | Krystel R Huxlin |

The funders had no role in study design, data collection and interpretation, or the decision to submit the work for publication.

### Author contributions
Sunwoo Kwon, Conceptualization, Data curation, Formal analysis, Methodology, Software, Validation, Visualization, Writing – original draft, Writing – review and editing; Berkeley K Fahrenthold, Matthew R Cavanaugh, Data curation, Writing – review and editing; Krystel R Huxlin, Conceptualization, Formal analysis, Funding acquisition, Methodology, Project administration, Resources, Supervision, Visualization, Writing – original draft, Writing – review and editing; Jude F Mitchell, Conceptualization, Data curation, Formal analysis, Funding acquisition, Methodology, Supervision, Validation, Visualization, Writing – original draft, Writing – review and editing

### Author ORCIDs
Sunwoo Kwon http://orcid.org/0000-0002-5761-6629
Krystel R Huxlin https://orcid.org/0000-0001-7138-6156
Jude F Mitchell https://orcid.org/0000-0003-0197-7545

### Ethics
All experimental protocols were conducted according to the guidelines of the Declaration of Helsinki and approved by The Research Subjects Review Board at the University of Rochester Medical Center (#00021951). Informed written consent was obtained from all participants prior to participation. Participants were compensated $15/hour.

### Decision letter and Author response
Decision letter https://doi.org/10.7554/eLife.67573.sa1
Author response https://doi.org/10.7554/eLife.67573.sa2

# Additional files

### Supplementary files
Transparent reporting form

### Data availability
Data for all figures has been shared on the Dryad. https://doi.org/10.6078/D1W69T.

The following dataset was generated:

| Author(s) | Year | Dataset title | Dataset URL | Database and Identifier |
|-----------|------|---------------|-------------|-------------------------|
| Kwon S, Farenthold BK, Cavanaugh MR, Mitchell JF, Huxlin KR | 2021 | Data from: Perceptual restoration fails to recover unconscious processing for smooth eye movements after occipital strokes | https://doi.org/10.6078/D1W69T | Dryad Digital Repository, 10.6078/D1W69T |

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
