## [Editor Report]

In this unique and meticulous study, Kwon, Huxlin and Mitchell add yet another twist to the story of dissociation between perception and action in the human brain. They found a sort of reverse blindsight: occipital stroke patients who lost conscious vision in part of the visual field could regain the ability to discriminate stimulus motion but did not make a specific form of eye movement that typically follows the stimulus motion. This result has implications for theories of perception and motor control, and for neurological rehabilitation.

---

## [Decision Letter]

**Decision letter after peer review:**

Thank you for submitting your article "Perceptual restoration fails to recover unconscious processing for smooth eye movements after occipital stroke" for consideration by *eLife*. Your article has been reviewed by 3 peer reviewers, and the evaluation has been overseen by a Miriam Spering as the Reviewing Editor and Chris Baker as the Senior Editor. The following individual involved in review of your submission has agreed to reveal their identity: Alex White (Reviewer #3).

The reviewers and reviewing editor have discussed the reviews with one another. The Reviewing Editor has drafted this letter to help you prepare a revised submission. In this letter, you will also find a summary of the post-review discussion.

Essential revisions:

As you will be able to read below, all reviewers appreciate the novelty of the question, the elegance of the study design, and the richness and high value of the obtained patient dataset. There is no question that the obtained results, should they indeed support the claimed dissociation, would be highly interesting to the vision sciences community and to the reader with a clinical background. However, as you will also see, all reviewers raised significant and major concerns with regard to data analysis and data interpretation. The reviewers share a concern that perhaps the issues cannot be addressed without collecting additional data, but – given the interest in the research question and value of the dataset – would like to give the authors a chance at rebuttal.

1. Throughout the paper, findings are interpreted to indicate a "clear dissociation between pathways mediating perceptual restoration and automatic actions", but the evidence for this dissociation is not clear and appears relatively overstated throughout. Specifically, this interpretation is based on a small subsample of the dataset (e.g., the regression of PRF gain on NDR thresholds in the intact field [Figure 4b] predicts that there should be a substantial PRF gain only at NDR thresholds below about 0.3. For the affected field this applies only to three data points of which one shows a substantial PFR and is fully compatible with the data in the intact field; there is also a concern about the ability to fit an exponential function here). In sum, the key finding of the paper appears to be based on an analysis that is drastically underpowered (for a theoretical coefficient r of 0.5, 80% power requires 29 subjects, far more than n=8).

In addition to recommendations listed below in individual reviews, two additional recommendations to potentially strengthen the finding of a dissociation were raised during the discussion: (a) the authors could add to Figure 4A a measure of the gain of post-saccadic ocular following for trials when the motion stimulus remains visible. On p 15, line 266-269 they make the "important observation" that the stroke patients are capable of making smooth following eye movements when fixating the moving stimulus. However, that effect has not been quantified. (b) Rather than relying on a correlation across subjects, the authors could run statistics on individual patient data and show that the PFR gain of at least some patients is significantly worse than would be predicted from their motion discrimination thresholds. For instance, they could use the data from the intact fields to build a model of how motion discrimination predicts PFR, and then test whether each data point from a blind field falls outside the confidence interval of that model.

In sum, without clear quantitative evidence for the claimed dissociation the overall impact of this paper would be relatively weak.

2. In terms of data analysis, the authors should report the mean values and confidence intervals for all quantitative measures (in addition to the reported statistics for the comparisons, exact p-values, t-values etc…). Importantly, the authors are encouraged to go beyond traditional null-hypothesis tests (t-tests and correlations) to make binary judgements of whether each effect or difference is "significant" (p<0.05). Some of the conclusions would be more convincing if supplemented with power analyses, bootstrapped confidence intervals, and Bayes factors to evaluate the strength of evidence.

3. Overall, the results might allow for a different interpretation: instead of assuming separate pathways for motion perception and oculomotor control in patients, the results might also be explained by a different read-out of the same motion signal for perception and oculomotor control, where oculomotor control applies a more conservative threshold and requires a higher internal signal strength than the motion perception. Further to the concern about data interpretation, a reviewer pointed out the potential role for presaccadic attentional mechanisms, which are currently not taken into account.

*Reviewer #1 (Recommendations for the authors):*

Unit of analysis: It seems that the unit of analysis was sometimes individual patients (N=8) and sometimes locations in the affected visual field (N=11). I think that treating multiple locations of one patient as independent samples is a form of pseudo replication (e.g. Lazic, 2010) and therefore the unit of analysis should always be individual patients.

Consciousness of motion perception: The authors interpret their motion discrimination task in terms of conscious motion perception, but strictly speaking the patients might also perform the task in a blindsight manner, where they have no conscious awareness of the motion itself but still are able to indicate the correct direction of motion.

Relevant literature: It might be interesting to relate the results to the theory and work by Milner and Goodale, who postulate a dissociation between perception and action as well.

Definition and values of NDR thresholds: I don't quite understand why the NDR thresholds in Table 1 and in Figure 4 range between 0 and 1 % although they should range between 0 and 100 % according to the equation in the methods.

Restoration training: It would be useful to provide some information in what kind of training the patients participated.

Reference

Lazic, S. E. (2010). The problem of pseudo replication in neuroscientific studies: is it affecting your analysis?. BMC neuroscience, 11(1), 1-17.

*Reviewer #2 (Recommendations for the authors):*

In order to comply as much as possible with the journal recommendations, the authors should report the mean values and confidence intervals for all quantitative measures (in addition to the reported statistics for the comparisons, exact p-values, t-values etc…)

*Reviewer #3 (Recommendations for the authors):*

(1) Throughout the manuscript, you rely on null-hypothesis tests that treat the outcome as binary, either significant if p<0.05, or completely supporting the null hypothesis if p>0.05. This is an outdated approach known to be problematic. For instance, consider the correlations in Figure 4B. One is test treated as significant and the other as showing no relationship. On line 244: "this relationship was lost in the blind field"; and line 276, "there was no significant correlation between restored NDR and PFR gains." If you take p>0.05 as your criterion for no relationship in a correlation, then yes. But the slope in Figure 4B doesn't appear to be 0. It seems quite possible that with more subjects from the same population, this correlation would be "significant." An example that goes in the other direction is in Supplementary Figure 2, the saccade angular deviations in the blind field: p=0.03, which is taken as strong evidence of normal motion perception in the blind field.

To improve your statistical analysis I would suggest several complementary approaches: (i) power analyses for "null" results to estimate how many participants would be required to get to p<0.05; (ii) Bayes Factors for the null vs alternate hypotheses. If you're using Matlab I recommend Bart Krekelberg's toolbox: https://klabhub.github.io/bayesFactor/ (ii) For one-sample t-tests I also recommend reporting 95% confidence intervals of the mean (obtained via bootstrapping), and for two-sample t-tests I would recommend reporting 95% bootstrapped confidence intervals for the mean difference.

(2) Some important details about the participants and the study design aren't explained until the Methods, which come at the end. Three details in particular confused me: (i) what the motion discrimination thresholds are, and how they're normalized. After I read the Methods section I better understood what the thresholds are, but I didn't understand the rationale for normalizing them the way you did. Why not just report the threshold direction range in degrees? (ii) The training that some patients underwent. It comes as a surprise in the Results that some visual field locations were trained and some weren't. Even in the abstract, the narrative is a little confusing. (iii) The two ways in which saccade "accuracy" is computed. One measure is reported in percent correct, and the other in degrees. You may just need to add a few words in the Results to clarify that first you categorized each saccade in a binary fashion as either landing near the target or near a different stimulus.

(3) A related question that should be clarified: when patients improve their motion discrimination after training, do they have a conscious percept of the motion, or is it another form of blindsight?

(4) I find these results to be very cool and indeed surprising. But I wonder whether they are limited to a particular set of unusual conditions, and a reader who spends less time thinking about oculomotor control may be puzzled by the particular set of conditions set up in this experiment. It does seem rather niche, in terms of the patient population, the particular eye movement, and the particular task and stimulus arrangement. Regarding the latter, your display contained 4 static apertures full of moving dots that don't really translate across the screen. Why did you need 4 apertures, rather than just one, and why not use single objects that actually moved? Does it matter that the target stimulus is 'competing' with three other irrelevant motion stimuli? Do these results generalize to more realistic settings when a person makes a saccade to a moving object that they then must pursue, like a bird flying across the sky? I myself have relied on such "unnatural" stimuli and I know there can be good reasons to use them carefully, but they need to be justified. And I imagine you had only a short amount of time with each patient, but we must also acknowledge that the whole story would be stronger if the phenomenon were demonstrated in a few different ways.

So overall, it would help if you could justify these peculiarities of your study and explain the ways in which they do and do not apply to other conditions – other people, other settings outside your particular lab experiment, and other behaviors.

---

## [Author Response]

Essential revisions:As you will be able to read below, all reviewers appreciate the novelty of the question, the elegance of the study design, and the richness and high value of the obtained patient dataset. There is no question that the obtained results, should they indeed support the claimed dissociation, would be highly interesting to the vision sciences community and to the reader with a clinical background. However, as you will also see, all reviewers raised significant and major concerns with regard to data analysis and data interpretation. The reviewers share a concern that perhaps the issues cannot be addressed without collecting additional data, but – given the interest in the research question and value of the dataset – would like to give the authors a chance at rebuttal.

We acknowledge the concern that the original sample size was not adequate and as such, we have collected additional data in 3 new patients in an attempt to address it.

1. Throughout the paper, findings are interpreted to indicate a "clear dissociation between pathways mediating perceptual restoration and automatic actions", but the evidence for this dissociation is not clear and appears relatively overstated throughout. Specifically, this interpretation is based on a small subsample of the dataset (e.g., the regression of PRF gain on NDR thresholds in the intact field [Figure 4b] predicts that there should be a substantial PRF gain only at NDR thresholds below about 0.3. For the affected field this applies only to three data points of which one shows a substantial PFR and is fully compatible with the data in the intact field; there is also a concern about the ability to fit an exponential function here). In sum, the key finding of the paper appears to be based on an analysis that is drastically underpowered (for a theoretical coefficient r of 0.5, 80% power requires 29 subjects, far more than n=8).

As described above, we have collected additional data from 3 new patients who attained training-induced recovery of global direction perception and integration below an NDR threshold of 0.35, the rough upper limit of the normal range. After several months required for recruiting, training and post-testing these patients, we were able to double the blind-field locations where normal performance was restored (from 3 to 6 independent locations). Although still a small sample size, the effects in the additional data remained highly consistent, showing no significant PFR. Using the combined data, there is still a significant difference in PFR between intact and blind-field locations under a variety of tests, including standard pairwise comparison tests, but also Bayes Factor t-tests as suggested by the reviewers. We have also repeated an analysis of the correlation between PFR and NDR threshold, but limited to the range of normal performance (NDR < 0.35) and find that the correlation remains non-significant for blind-field locations in that range (R^2^=0.362, F_4_=2.267, p=0.2066, BF=0.6993). However, as the reviewers anticipated, even with the added patients (now n=6) testing the difference in slopes for those line fits in the correlation would require a much larger n, and the slopes fit in this range are not significantly different (slopes of -0.8616 vs -0.5611, t_22_=-0.6478, p=0.5238). By contrast, if slopes are compared over the full NDR range, including all points, they are significantly different (slopes -0.8616 vs -0.0385, t_30_=-2.9461, p=0.0062). Given the difficulty obtaining the 3 additional patients who recovered back into the normal NDR range, we have instead adopted the approach suggested by Reviewer 1 for this issue: we fit an exponential curve to the intact-field points and evaluated if the PFR for blind-fields fall outside that distribution. For that analysis, we did find robust support for a significant difference, as detailed below in our responses to Reviewer 1. We have updated Results to include the additional statistics for the restricted range and the exponential fit analyses on lines 310-323.

In addition to recommendations listed below in individual reviews, two additional recommendations to potentially strengthen the finding of a dissociation were raised during the discussion: (a) the authors could add to Figure 4A a measure of the gain of post-saccadic ocular following for trials when the motion stimulus remains visible. On p 15, line 266-269 they make the "important observation" that the stroke patients are capable of making smooth following eye movements when fixating the moving stimulus. However, that effect has not been quantified.

We now provide quantification of occipital stroke patients’ smooth following eye movements when motion remains visible at saccade end. Velocity traces after saccade landing exhibit smooth following of foveal stimulus motion, which for blind-fields occurs with a roughly 100 ms visuomotor delay (Figure 3D, red curve) as compared to intact-fields, where pre-saccadic information influences an earlier predictive response (Figure 3C, red curve). The quantified PFR gain and updated analyses are now presented in Results (lines 290-295, Figure 3—figure supplement 1). CB patients have no problem generating smooth, following eye movements for stimulus motion at the fovea after a visual response delay. It is only the predictive “open-loop” period (from 20-100 ms post-saccade) that differs between intact and blind-fields. That early period relies on motion processing from peripheral locations before the saccade is initiated, and this remains the focus of our main results.

(b) Rather than relying on a correlation across subjects, the authors could run statistics on individual patient data and show that the PFR gain of at least some patients is significantly worse than would be predicted from their motion discrimination thresholds. For instance, they could use the data from the intact fields to build a model of how motion discrimination predicts PFR, and then test whether each data point from a blind field falls outside the confidence interval of that model.In sum, without clear quantitative evidence for the claimed dissociation the overall impact of this paper would be relatively weak.

We now show results of an exponential fit to the intact visual field data to describe the dependence of PFR on motion integration performance as a supplemental analysis (Figure 4—figure supplement 1). In this figure, the solid red line represents the model fit based on the intact-field data. The dotted red lines represent the confidence interval boundaries at 95%. Out of a total of 6 blind-field points (including data from our newly-recruited patients) within the normal range of NDR performance in the intact-field (white data points), 5 lie outside these confidence intervals, and have individual confidence intervals on their PFR gain that are instead overlapping or below zero. The probability that 5 of 6 points would lie outside the 95% confidence intervals, assuming they came from the same distribution as intact-fields is highly improbable (p <= 1.79x10^-6^, binomial test). We also note that the single point which fell within the range of the intact visual fields was unusual in other ways. Unlike other blind-field locations in our study, this point (Figure 1, CB6, lower left quadrant) did exhibit significant detection performance in Humphrey visual field tests (spared Humphrey visual field performance – i.e. luminance detection), even though global motion discrimination at that location was impaired and thus selected for training (see Methods). Nonetheless, even when this potential outlier was included in our analysis, we still found clear quantitative evidence to support the distinction in PFR gain within the range where normal motion discrimination performance has been recovered. This new analysis is included in Results (lines 310-323) and with Figure 4—figure supplement 1.

2. In terms of data analysis, the authors should report the mean values and confidence intervals for all quantitative measures (in addition to the reported statistics for the comparisons, exact p-values, t-values etc…).

The manuscript has now been modified to include these additional values throughout.

Importantly, the authors are encouraged to go beyond traditional null-hypothesis tests (t-tests and correlations) to make binary judgements of whether each effect or difference is "significant" (p<0.05). Some of the conclusions would be more convincing if supplemented with power analyses, bootstrapped confidence intervals, and Bayes factors to evaluate the strength of evidence.

We have expanded our analyses to provide boot-strapped confidence intervals for PFR across subjects (Figure 4A). In addition, we now include Bayes Factor t-tests for all the pair-wise comparisons in the revised manuscript.

3. Overall, the results might allow for a different interpretation: instead of assuming separate pathways for motion perception and oculomotor control in patients, the results might also be explained by a different read-out of the same motion signal for perception and oculomotor control, where oculomotor control applies a more conservative threshold and requires a higher internal signal strength than the motion perception.

This alternative interpretation merits consideration – we have expanded the Discussion to include it by adding a new paragraph (lines 393-408).

Further to the concern about data interpretation, a reviewer pointed out the potential role for presaccadic attentional mechanisms, which are currently not taken into account.

We considered the possibility that impairments of attentional mechanisms might impact PFR in our original manuscript (see original Discussion), but dismissed it as highly unlikely for two reasons: first, prior literature shows that attention can be directed into blind-fields and improve performance (Poggel et al., 2006; Kentridge et al., 2004; Cavanaugh et al., 2019). Second, we also find in the current paradigm, evidence for attentional selection of target motion in the deviation of saccade end-points along the motion direction of the target. We quantified this effect by measuring the angular deviations of saccades along target motion (the angle from the fixation point to the target center and to the saccade end-point). Saccades were positively deviated, reflecting pre-saccadic selection of target motion for both the intact and blind-fields (Figure 4—figure supplement 4). Similar shifts in saccade end-point have been correlated with changes in the perception of target location in previous studies (Schafer and Moore, 2007; Kosovicheva et al., 2014; Kwon et al., 2019), and may be more tightly correlated with perception than are smooth eye movements. Thus, we do find some positive evidence for attentional selection, although saccade deviations were weaker for blind that intact-fields, which could reflect sensory or attention deficits, and thus we sought additional controls.

To further test for deficits in attention allocation, we ran a new, control experiment in 2 of our new CB participants and implemented an analysis suggested by both Reviewers 2 and 3. These additional results appear in the Discussion where we expand the previous section on the role of attention (lines 442-462). In short, we asked if the lack of PFR in the blind-field was caused by an inability to allocate pre-saccadic attention to a blind-field target, against the competing influence of distractor apertures in intact regions of the visual field. We found no measurable PFR in the blindfield, even when distractor motion was removed, and PFR remained robust in the intact-fields comparable to the original design (Figure 4—figure supplement 5B). These additional results, in combination with the original positive result for saccade end-points, support that the lack of PFR does not reflect an impairment to allocate attention to the pre-saccadic target location.

Reviewer #1 (Recommendations for the authors):Unit of analysis: It seems that the unit of analysis was sometimes individual patients (N=8) and sometimes locations in the affected visual field (N=11). I think that treating multiple locations of one patient as independent samples is a form of pseudo replication (e.g. Lazic, 2010) and therefore the unit of analysis should always be individual patients.

We understand the reviewer’s concern. However, while one may claim inter-dependence of visual field locations within a given participant when dealing with visually-intact participants, there is considerable evidence in the CB literature to show that chronic cortically-blinded fields do not behave normally in this respect (Sahraie et al., 2006; Huxlin et al., 2009; Das et al., 2014; Saionz et al., 2020). Baseline performance and the ability of a location to improve are not uniform across CB fields, even at identical eccentricities. More relevant to the present comparisons, training-induced changes in discrimination and motion integration performance in the blind-field are tightly restricted to the trained, blind-field location and do not spread by more than 1° outside the boundaries of each trained location (Huxlin et al., 2009; Das et al., 2014; Saionz et al., 2020). This is unique – to our knowledge – to cortically-blinded fields and motivates why we maintain that it is appropriate to treat different blind-field locations as independent sites for training assessments. Nonetheless, we now present subject data in all figures except for Figure 4B and 4C, where we compare NDR thresholds and PFR for individual intact (n=20) and blind (n=14) field locations. In addition, we have included additional figures (Figure 4—figure supplement 2A and 2B) to show the outcome of re-analyzing the latter data set while pooling results per CB patient (11 patients = 11 intact vs. 11 blind-fields). Even when pooling data per subject, we still observe the trends shown in Figure 4B and 4C: lower NDR is correlated with higher PFR gain in the intact-fields (R^2^=0.568, F_9_=11.8334, p=0.0074, BF=8.0855), and no significant correlation between NDR and PFR gain in the blind-fields (R^2^=0.038, F_9_=0.3552, p=0.5659, BF=2.650) across the entire range of NDR. Further, for NDR < 0.35, the correlation between PFR and performance remains significant for intact-fields (R^2^=0.568, F_9_=11.8334, p=0.0074, BF=8.0855) but not for blindfields (R^2^=0.5406, F_2_=2.3538, p=0.2647, BF=0.7678). When we compared whether the slopes from each linear regression with respect to intact-fields and blind-fields under NDR < 0.35, we found that slopes (-1.012 for intact and -0.1845 for blind) were significantly different from each other (t_11_=-2.6039, p=0.0245). Thus, our main findings, where NDR thresholds and PFR gains were correlated per location, still hold when locations are pooled per subject. We have added this information in the revised manuscript’s Results section (lines 324-332).

Consciousness of motion perception: The authors interpret their motion discrimination task in terms of conscious motion perception, but strictly speaking the patients might also perform the task in a blindsight manner, where they have no conscious awareness of the motion itself but still are able to indicate the correct direction of motion.

We thank the reviewer for these suggestions. With respect to the first point, the Huxlin lab has previously described the properties of visual perception associated with training-induced recovery of NDR thresholds in cortically-blinded fields (a few key papers include Huxlin et al., 2009; Das et al., 2014; Saionz et al., 2020). In particular, the paper by Das and colleagues illustrated how direction integration of motion information presented by our random dot stimuli is not capable of eliciting blindsight. That global motion does not elicit blindsight was also shown by Azzopardi and Cowey (2001) and Vaina et al. (2014) among others. Prior to training, our patients were unable to describe or even correctly guess global motion direction in these stimuli in their blind-field – as such, they had no demonstrable blindsight for our global direction integration task. By the end of training, they could both correctly describe their conscious percept of the random dot stimuli and attain normal/near-normal NDR threshold levels of performance. The fact that patients are fully aware and can describe the stimulus in detail as well as how that percept changes when direction range within the stimulus is increased or decreased at different steps of the staircase within a session further implies conscious perception and not blindsight. We have added this information to the text of the Discussion (lines 525532).

Relevant literature: It might be interesting to relate the results to the theory and work by Milner and Goodale, who postulate a dissociation between perception and action as well.

We agree that perceptual therapies which restore conscious vision in CB appear to exhibit a dissociation with processing for action that naturally leads to parallels with the work of Milner and Goodale (TINS, 1992) where it was suggested that ventral stream projections from the striate cortex to the inferotemporal cortex play a major role in perception, while dorsal stream projections from the striate cortex to posterior parietal regions mediates the required sensorimotor transformations for visually-guided actions. Our current results suggest that after V1 damage, psychophysical training enables residual motion processing, likely in area MT and related dorsal pathway areas, to be used to recover perception without recovering the necessary processing for smooth eye movements. This suggests a dissociation between perception and visually-guided actions in this rehabilitation paradigm. We have added this information to the text of the Discussion (lines 532-541).

Definition and values of NDR thresholds: I don't quite understand why the NDR thresholds in Table 1 and in Figure 4 range between 0 and 1 % although they should range between 0 and 100 % according to the equation in the methods.

Our apologies – we fixed the equation in the method to reflect that NDR thresholds should range from 0 to 1 instead of 0 to 100%.

Restoration training: It would be useful to provide some information in what kind of training the patients participated.Reference: Lazic, S. E. (2010). The problem of pseudo replication in neuroscientific studies: is it affecting your analysis?. BMC neuroscience, 11(1), 1-17.

We added some information about the training administered to participants in the

Methods section, along with references in which training protocols are described in even more detail (lines 584-607). We address the Lazic reference as described above with a supplemental analysis done per subject (Figure 4—figure supplement 2).

Reviewer #2 (Recommendations for the authors):In order to comply as much as possible with the journal recommendations, the authors should report the mean values and confidence intervals for all quantitative measures (in addition to the reported statistics for the comparisons, exact p-values, t-values etc…)

We have revised the manuscript to include the requested values in addition to the Bayes Factor analysis.

Reviewer #3 (Recommendations for the authors):(1) Throughout the manuscript, you rely on null-hypothesis tests that treat the outcome as binary, either significant if p<0.05, or completely supporting the null hypothesis if p>0.05. This is an outdated approach known to be problematic. For instance, consider the correlations in Figure 4B. One is test treated as significant and the other as showing no relationship. On line 244: "this relationship was lost in the blind field"; and line 276, "there was no significant correlation between restored NDR and PFR gains." If you take p>0.05 as your criterion for no relationship in a correlation, then yes. But the slope in Figure 4B doesn't appear to be 0. It seems quite possible that with more subjects from the same population, this correlation would be "significant." An example that goes in the other direction is in Supplementary Figure 2, the saccade angular deviations in the blind field: p=0.03, which is taken as strong evidence of normal motion perception in the blind field.

The conclusions regarding Figure 4B (now split into Figures 4B and 4C) have been substantially strengthened by collecting additional data and performing requested analyses (see above, revisions in Results on lines 303-323). We have also revised the Discussion to consider other supporting data and alternatives with less binary outcomes.

To improve your statistical analysis I would suggest several complementary approaches: (i) power analyses for "null" results to estimate how many participants would be required to get to p<0.05; (ii) Bayes Factors for the null vs alternate hypotheses. If you're using Matlab I recommend Bart Krekelberg's toolbox: https://klabhub.github.io/bayesFactor/ (ii) For one-sample t-tests I also recommend reporting 95% confidence intervals of the mean (obtained via bootstrapping), and for two-sample t-tests I would recommend reporting 95% bootstrapped confidence intervals for the mean difference.

Other reviewers also suggested the inclusion of Bayes Factors and boot-strapped confidence intervals, which we have now implemented throughout the manuscript. We find that all the results initially reported stand using these tests and the additional data we have collected.

(2) Some important details about the participants and the study design aren't explained until the Methods, which come at the end. Three details in particular confused me: (i) what the motion discrimination thresholds are, and how they're normalized. After I read the Methods section I better understood what the thresholds are, but I didn't understand the rationale for normalizing them the way you did. Why not just report the threshold direction range in degrees?

We have revised the Results to define how NDR thresholds are computed when they are first introduced (lines 127-135). The Huxlin and Pasternak labs have used direction range measures extensively in their research on motion processing in humans and animal models with cortical damage. The reason direction range in degrees has increasingly been converted to NDR thresholds in more recent publications from both labs is because direction range scales from low to high values as performance improves, and this is the inverse of the scaling for thresholds (where lower values mean better performance). NDR is a simple means of aligning integration performance to standards for reporting thresholds in the field.

(ii) The training that some patients underwent. It comes as a surprise in the Results that some visual field locations were trained and some weren't. Even in the abstract, the narrative is a little confusing.

We apologize for the confusion and have now clarified (Results: lines 117-126; Methods: lines 591-622) that our main goal was to assess whether and how PFR tracked with perception in the blind- and intact-fields. As such, PFR testing locations (which had to be within a specific eccentricity range and mirror symmetric across the 4 quadrants of the visual field) did not always fall on a trained, blind-field location (the requirements for selecting training locations are quite different than those for PFR measurements). This did not matter at all. In fact, the tight location and task-specificity of training-induced NDR recovery in the blind-field allowed us to attain a large range of NDR thresholds at blind-field PFR test locations, against which we were able to plot PFR gain.

(iii) The two ways in which saccade "accuracy" is computed. One measure is reported in percent correct, and the other in degrees. You may just need to add a few words in the Results to clarify that first you categorized each saccade in a binary fashion as either landing near the target or near a different stimulus.

We thank the reviewer for noting this point – we have clarified the two different measures of saccade accuracy where they are first introduced in Results (lines 174-178).

(3) A related question that should be clarified: when patients improve their motion discrimination after training, do they have a conscious percept of the motion, or is it another form of blindsight?

Please see our earlier response to Reviewer 1, and new information added to the Discussion (lines 525-532). In sum, our data support recovery of conscious perception and not just blindsight.

(4) I find these results to be very cool and indeed surprising. But I wonder whether they are limited to a particular set of unusual conditions, and a reader who spends less time thinking about oculomotor control may be puzzled by the particular set of conditions set up in this experiment. It does seem rather niche, in terms of the patient population, the particular eye movement, and the particular task and stimulus arrangement. Regarding the latter, your display contained 4 static apertures full of moving dots that don't really translate across the screen. Why did you need 4 apertures, rather than just one, and why not use single objects that actually moved? Does it matter that the target stimulus is 'competing' with three other irrelevant motion stimuli?

A key advantage of the current paradigm is that it enables one to measure the postsaccadic following response (PFR) to a motion stimulus that is spatially localized, and at the same time, enables one to plan a saccade to a specific spatial location, even if perception at that location is impaired. The paradigm was adapted from an earlier study (Kwon et al., 2019) which examined the effects of pre-saccadic attention when saccades were cued by a central line at fixation towards one of 4 apertures. We retained that design for consistency with the previous study (allowing comparison between patients and previously-tested controls) and because of the control it provides both in directing the saccade and localizing motion to a specific visual field location. However, we agree that it is not critical to have four apertures to obtain similar spatial localization and spatial cueing for saccades. As such, we ran an additional control experiment in two new patients using a single stimulus (Figure 4—figure supplement 5) and we were able to validate similar results regardless of whether one or four apertures were used. This suggest that observed effects do not rely on the complexity of the 4-stimulus apertures, or complexities in how attention is allocated between them, as discussed earlier under “Essential revisions for authors”, which is included in Discussion (lines 442-454).

Do these results generalize to more realistic settings when a person makes a saccade to a moving object that they then must pursue, like a bird flying across the sky? I myself have relied on such "unnatural" stimuli and I know there can be good reasons to use them carefully, but they need to be justified. And I imagine you had only a short amount of time with each patient, but we must also acknowledge that the whole story would be stronger if the phenomenon were demonstrated in a few different ways.

The rationale for our stimulus design is outlined in our response above, but as noted the findings do appear to generalize if a single aperture is used, and we believe they would generalize further to natural stimuli without the control afforded by using an aperture. If anything, the controlled conditions afforded by presenting stimuli within an aperture would limit our ability to measure the PFR and provide conservative estimates of its magnitude under natural conditions.